# ADIOS: Antibody Development via Opponent Shaping

**Sebastian Towers** [* 1 2]   **Aleksandra Kalisz** [* 1 2]   **Philippe A. Robert** [3]   **Alicia Higueruelo** [4]   **Francesca Vianello** [5]
**Ming-Han Chloe Tsai** [6]   **Harrison Steel** [2]   **Jakob Foerster** [1 2]

## Abstract

Anti-viral therapies are typically designed to target only the current strains of a virus, a *myopic* response. However, therapy-induced selective pressures drive the emergence of new viral strains, against which the original myopic therapies are no longer effective. This evolutionary response presents an opportunity: our therapies could both *defend against and actively influence viral evolution*. This motivates our method ADIOS: Antibody Development vIa Opponent Shaping. ADIOS is a meta-learning framework where the process of antibody therapy design, the *outer loop*, accounts for the virus's adaptive response, the *inner loop*. With ADIOS, antibodies are not only robust against potential future variants, they also influence, i.e., *shape*, which future variants emerge. In line with the opponent shaping literature, we refer to our optimised antibodies as *shapers*. To demonstrate the value of ADIOS, we build a viral evolution simulator using the Absolut! framework, in which shapers successfully target both current and future viral variants, outperforming myopic antibodies. Furthermore, we show that shapers modify the distribution over viral evolutionary trajectories to result in weaker variants. We believe that our ADIOS paradigm will facilitate the discovery of long-lived vaccines and antibody therapies while also generalising to other domains. Specifically, domains such as antimicrobial resistance, cancer treatment, and others with evolutionarily adaptive opponents. Our code is available at `https://github.com/olakalisz/adios`.

*Equal contribution [1]FLAIR, Foerster Lab for AI Research [2]Department of Engineering, University of Oxford, Oxford, UK [3]Department of Biomedicine, University of Basel, Basel, Switzerland [4]Isomorphic Labs, London, UK [5]Exscientia, Oxford, UK [6]Epsilogen Ltd., London, UK. Correspondence to: Sebastian Towers <sebastian.towers@eng.ox.ac.uk>, Aleksandra Kalisz <aleksandra.kalisz@eng.ox.ac.uk>.

*Proceedings of the 42nd International Conference on Machine Learning*, Vancouver, Canada. PMLR 267, 2025. Copyright 2025 by the author(s).

## 1. Introduction

Designing effective therapies to fight off viral pathogens is crucial for limiting their devastating social and economic costs (Nandi & Shet; Orenstein & Ahmed, 2017; Samsudin et al., 2024; Faramarzi et al., 2024). However, traditional design approaches only target the *current* variant of a virus. Although this *myopic* design approach may yield therapies with high initial efficacy, it fails to account for viral adaptation, leaving treatments vulnerable to becoming ineffective over time (Weisblum et al., 2020; Doud et al., 2018; Lee et al., 2019; Dingens et al., 2019; Greaney et al., 2021).

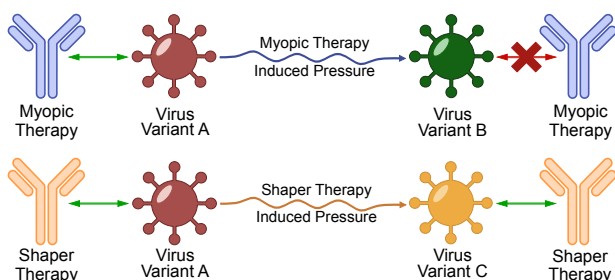

*Figure 1.* **Myopic Therapies Become Ineffective Over Time.** Initial virus (variant A) evolves in response to the evolutionary pressures induced by therapies, resulting in new variants. In case of traditional *myopic* therapies (top), the new emerging variants (variant B) are often therapy resistant. In contrast, ADIOS designs *shapers* or shaper therapies (bottom) which remain effective and steer the viral evolution towards less harmful variants (variant C).

The COVID-19 pandemic starkly illustrated the challenges of adaptive viruses. While the rapid development of vaccines was a remarkable achievement, concerns quickly arose about their long-term efficacy against new emerging COVID variants (Carabelli et al., 2023; Hu et al., 2021). For example, the B.1.351 variant demonstrated that the vaccine loses its effectiveness against new strains (Madhi et al., 2021). This underscores the need for approaches that consider both the current *and future* efficacy of a designed therapy.

The virus inevitably adapts in response to selective pressures imposed by our therapies, i.e., we *influence* the viral evolution (Chéron et al., 2016; Meijers et al., 2022). Our work turns this influence in our favour, designing therapies that steer the virus towards less dangerous variants, see Figure 1.

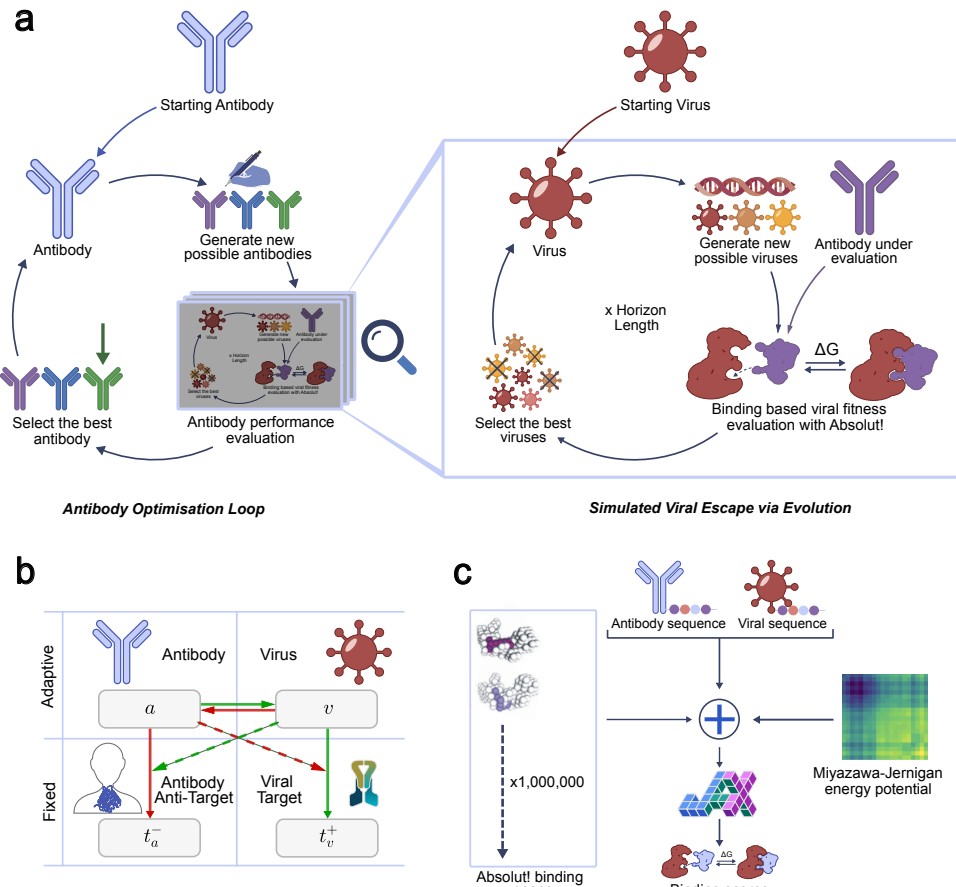

*Figure 2.* **Main Components of ADIOS. a** The ADIOS framework. In the *Antibody Optimisation Loop* (i.e., *outer loop*), we optimise the antibody to perform well against current and future virus variants; thus influencing the viral evolution. We approximate the future variants through our *Simulated Viral Escape via Evolution* (i.e., *inner loop*) where the viruses evolve to escape from the current antibody over a given horizon length. **b** The payoffs of the antibody and virus. Red arrows indicate binding interactions that players aim to minimise, while green arrows represent those they aim to maximise. The antibody optimises for binding to the virus while avoiding its anti-target. In this zero-sum game, the antibody's optimisation indirectly counters the virus's binding to its target, see Equation 1. **c** Binding simulator. Our JAX (Bradbury et al., 2018) implementation of the binding calculation uses binding poses generated by Absolut! (Robert et al., 2022) and the Miyazawa-Jernigan energy potential matrix (Miyazawa & Jernigan, 1999).

To achieve this we utilise principles from *opponent shaping* (Foerster et al., 2018), a multi-agent reinforcement learning framework that allows agents to both *anticipate* and *influence* the future policies of other agents in their environment. This approach, exemplified by methods such as Learning with Opponent-Learning Awareness (Foerster et al., 2018) and Model-Free Opponent Shaping (Lu et al., 2022), allows agents to consider not only their current performance but also the consequences of their actions on their opponents' future behaviour.

Building on these principles, we introduce ADIOS: Antibody Development vIa Opponent Shaping. Antibodies are immune system proteins that bind to pathogens such as viruses. While naturally produced by the body, it is also possible to design and synthetically produce antibodies as therapies. ADIOS frames the interaction between antibodies and viruses as a *two-player zero-sum game*. In this game,

the antibody's payoff is primarily determined by its binding strength to the virus, while the virus has the opposite payoff (Figure 2b). Although our framework can use any binding model in principle, in this work we build on the Absolut! framework (Robert et al., 2022) to estimate the binding strength of protein-protein interactions. To improve computational efficiency, we reimplement parts of Absolut! in JAX (Bradbury et al., 2018), allowing GPU acceleration and a 10,000-fold speedup over the original implementation (Figure 2c).

We use this game to model *viral escape* - the process through which mutations allow a virus to evade a host's immune system (Lucas et al., 2001). Following a meta-learning approach, ADIOS implements two nested optimisation loops, an *inner loop* and an *outer loop* (Figure 2a). In the inner loop, we simulate viral escape via evolution, where the virus adapts to the current antibody by repeatedly finding approx-

imate best responses that decrease binding strength. The outer loop uses a genetic algorithm to optimise the antibodies to be effective across viral evolutionary trajectories, resulting in antibodies we call *shapers*. This is in contrast to only optimising for binding to the initial virus, which results in *myopic* antibodies.

Importantly, our simulations show that shapers not only outperform myopic antibodies in long-term efficacy but also demonstrate the ability to shape viral evolution. Moreover, shapers guide viruses toward variants that are more susceptible to binding to a broad spectrum of antibodies, not just the shaper that induced the given viral evolution, providing insights into the scalability and potential for practical deployment of ADIOS. Our study also explores the trade-offs between the effectiveness of shapers and the computational resources required for their optimisation. Finally, we present an explainability analysis of the key features that distinguish shapers from myopic antibodies.

Our key contributions include:

- ADIOS: A framework that brings opponent shaping to antibody design to address viral escape.
- A GPU-accelerated JAX implementation of the binding simulator Absolut!, achieving a 10,000x speedup.
- An open-source instantiation of ADIOS applied to antibody design for both the dengue virus and three other viruses using our JAX implementation.
- Empirical results showing ADIOS-optimised shapers both significantly outperform myopic antibodies by limiting long-horizon viral escape and guide viral evolution towards variants that can be more easily targeted.
- Analysis of computational trade-offs in shaping horizons, providing practical guidance for deploying ADIOS in compute-constrained settings, e.g. due to more realistic binding simulators.
- Interpretability analysis into how antibody shapers influence viral escape, which could, in principle, provide inspiration to antibody designers.

While our results provide a promising proof of concept, they are based on simplified models of binding and viral escape. However, we believe that as more sophisticated simulators emerge, the ADIOS framework has the potential to significantly impact future antiviral therapy design.

## 2. Related Work

**Antibody Design:** Antibodies are essential components of the immune system that bind to unique identifiers (antigens) present on pathogens, including viruses, to identify and neutralise them. While natural antibodies emerge through an immune response, it is also possible to *design* antibodies for use as therapies. Recent work has made significant progress in computational antibody design (Cutting et al., 2024; Zam-

baldi et al., 2024; Bennett et al., 2024). The common approaches to antibody design utilise energy-based antibody optimisation methods (Li et al., 2014; Adolf-Bryfogle et al., 2018; Pereira et al., 2024), sequence-based language models (Liu et al., 2020; Saka et al., 2021) or structure-based approaches relying on GNNs (Jin et al., 2022) and diffusion models (Martinkus et al., 2024).

In contrast to these works, we are not interested in generating better antibody design methods immediately, but rather in how we should make new methods in the future to account for our effect on evolving viruses.

**Predicting Viral Escape:** Recent machine learning methods have demonstrated success in predicting future viral strains (Shanker et al., 2024; Wang et al., 2023; Nie et al., 2025). EVEscape (Thadani et al., 2023) decomposes the likelihood of a mutation into three parts: maintaining fitness, accessibility to antibodies, and disrupting binding, demonstrating success through retrospective identification of COVID variants. Han et al. (2023) take a different approach by modelling viral evolution through simulated fitness landscapes. Unlike these methods, ADIOS models the antibody influence on viral evolution, enabling both the simulation of viral escape trajectories and the optimisation of antibodies to minimise viral escape.

## 3. Background

**Antibody Binding Simulators:** In our setting, the interaction between antibodies and viruses is characterised by their binding strength $B(\cdot, \cdot)$ - a measure of how strongly the two "attach" to each other through molecular forces. Molecular dynamics simulations offer high accuracy but are computationally intensive (Hollingsworth & Dror, 2018). Sequence-based ML models (Mason et al., 2021; Lim et al., 2022; Ruffolo et al., 2023; Yan Huang et al., 2022) provide faster alternatives but struggle to generalise beyond their training distribution, making them unsuitable for exploring novel viral mutations. To evaluate $B(\cdot, \cdot)$ we use the Absolut! framework (Robert et al., 2022). Absolut! offers a balance between speed and generalisation by modelling binding through discretised protein structures. It focuses on the CDRH3 region of the antibody, the most variable portion that primarily determines binding specificity (VanDyk & Meek, 1992). For each antibody-antigen pair, Absolut! enumerates possible binding poses and computes their energy using the Miyazawa-Jernigan potential (Miyazawa & Jernigan, 1999). The binding strength $B(\cdot, \cdot)$ is then defined as the negative of the lowest binding energy, see Figure 2c and Appendix D for details.

**Opponent Shaping:** A multi-agent reinforcement learning framework which allows agents to anticipate and influence the future policies of other agents in their environment. Learning with Opponent-Learning Awareness (LOLA) (Foerster et al., 2018) introduced this concept by having LOLA

**Algorithm 1** $\text{Ev}(\hat{v}, a)$: Simulated Viral Escape (Inner Loop)

> **Input:** virus $\hat{v}$, antibody $a$, horizon $H$, population size $P$, inverse temperature $\beta$
> **Output:** Sampled viral trajectory $\hat{\boldsymbol{v}} = [\hat{v}^0, \hat{v}^1, \ldots, \hat{v}^H]$
> $\hat{v}_0 \leftarrow \hat{v}$
> **for** $i = 0$ **to** $H - 1$ **do**
>    **for** $k = 1$ **to** $P$ **do**
>       $v_k^i \leftarrow \hat{v}^i \oplus \textit{Mutation}$
>    **end for**
>    $p(v) \leftarrow \mathbb{P}(v = v_k^i) \propto \exp(\beta R_\text{v}(v_k^i, a))$     // See Eq. 1
>    $\hat{v}^{i+1} \sim p(v)$
> **end for**
> $\hat{\boldsymbol{v}} \leftarrow [\hat{v}_0, \ldots, \hat{v}_H]$
> **return** $\hat{v}$

**Algorithm 2** Antibody Optimisation (Outer Loop)

> **Input:** antibody $\hat{a}$, virus $\hat{v}$, horizon $H$, population size $P_a$, steps $N$
> **Output:** Trajectory of antibodies $\hat{\boldsymbol{a}} = [\hat{a}^0, \hat{a}^1, \ldots, \hat{a}^N]$
> $\hat{a}_0 \leftarrow \hat{a}$
> **for** $i = 0$ **to** $N - 1$ **do**
>    $a_1^i \leftarrow \hat{a}^i$
>    **for** $k = 2$ **to** $P_a$ **do**
>       $a_k^i \leftarrow \hat{a}^i \oplus \textit{Point Mutation}$
>    **end for**
>    $\hat{a}^{i+1} \leftarrow \arg\max_k \mathbb{E}\left[ F_{\hat{v}}^H(a_k^i) \right]$     // See Eq. 2
> **end for**
> $\hat{\boldsymbol{a}} \leftarrow [\hat{a}_0, \ldots, \hat{a}_N]$
> **return** $\hat{a}$

agents optimise against anticipated opponent updates rather than static opponent policies. They achieved this through an augmented value function that accounts for the opponent's learning step.

Meta-learning is a set of methods for optimising a learning process itself, "learning to learn". In multi-agent systems, this concept extends to learning about and influencing how other agents learn. Model-Free Opponent Shaping (M-FOS) (Lu et al., 2022) showcases this idea by using gradient-free optimisation to learn meta-policies that accomplish long-horizon opponent shaping. Our approach follows a similar principle, using evolutionary optimisation to shape viral escape trajectories.

## 4. Method

ADIOS frames antibody design as a two-player game between an antibody shaper agent and a naive virus agent, building on principles from opponent shaping (Figure 2). We present our method in three parts:

In Section 4.1, we introduce the **virus-antibody game**, defining the action spaces and payoffs for both players. Section 4.2 describes our **simulated viral evolution** process, modelling how viruses evolve to escape a given antibody. Finally, Section 4.3 presents our **antibody optimisation** approach, which optimises the antibody shapers in a way that accounts for *future viral mutations* and learns to *influence* viral evolution away from escape.

### 4.1. Virus-Antibody Game

We formalise the interaction between antibodies and viruses as a *two-player zero-sum game*. In this game, two players – the virus and the antibody – play a game where one player's gain is the other's loss. The game is defined by the set of actions available to each player and their respective payoffs. The players' actions are represented by their amino acid sequences. The sequences are of an antigen protein for the virus and a fragment of a hypervariable region of the heavy chain for the antibody.

We define the set of 20 amino acids as $\mathbb{A}$. Let $N_v$ be the virus sequence length and $N_a$ be the antibody sequence length. So an action of the virus is $v \in \mathbb{A}^{N_v}$, and an action of the antibody is $a \in \mathbb{A}^{N_a}$. Let $B : \mathbb{A}^{N_v} \times \mathbb{A}^{N_a} \to \mathbb{R}$ be our binding function, which measures the strength of the binding between the antibody and the virus with increasing values corresponding to stronger binding[1].

The payoff structure is designed to capture the biological incentives of both players: the antibody aims to bind strongly to the virus while avoiding binding to human proteins (an *anti-target*), whereas the virus seeks to evade antibody binding while maintaining its ability to bind to host cell receptors (a *binding target*). Mathematically, we define the antibody's payoff $R_a$ as:

$$R_a(v, a) = B(v, a) - B(t_a^-, a) - B(v, t_v^+) \qquad (1)$$

where $B(v, a)$ represents the binding strength between the virus $v$ and antibody $a$, $t_a^-$ is the antibody's anti-target, and $t_v^+$ is the virus's binding target. The virus's payoff $R_v$ is simply the negative of the antibody's payoff: $R_v(v, a) = -R_a(v, a)$, see Figure 2b. This formulation also ensures that neither player can adopt an overly simplistic strategy: the virus can't become entirely inert without losing its ability to infect host cells, and the antibody can't become universally "sticky" without binding to the human protein, a "false positive", potentially causing the immune system to attack the human body. We give the full Markov Decision Process (MDP) definition $\mathcal{M} = \langle \mathcal{S}, \mathcal{A}^\text{v}, \mathcal{A}^\text{a}, P, R, \mu \rangle$ in Appendix E.

### 4.2. Simulated Viral Escape via Evolution

We model the viral escape as a virus $\hat{v}$ naively evolving for $H$ steps in response to some fixed antibody $a$. The *simulated viral escape via evolution*, see Figure 2a and Algorithm 1, is defined as follows. Given a starting virus $\hat{v}$, the fixed antibody $a$ induces a distribution $\text{Ev}(\hat{v}, a)$ over sequences

---

[1]This is opposite to binding *energies*, which are smaller for stronger binding.

of viruses $\hat{\boldsymbol{v}} = [\hat{v}^0, \hat{v}^1, \hat{v}^2, \ldots \hat{v}^H]$, where $\hat{v}^0 = \hat{v}$ and $H$ is the chosen horizon length. We write $\hat{\boldsymbol{v}} \sim \text{Ev}(\hat{v}, a)$ to denote this relationship.

We define the process of generating the escape trajectories inductively. In generation $i$, we have a virus $\hat{v}^i$. We generate a population of viruses $v_1^i, v_2^i \ldots v_P^i$ by duplicating $\hat{v}^i$ $P$ times, then randomly applying mutations such that on average there is one amino acid mutation per viral sequence:

$$v_k^i = \hat{v}^i \oplus \textit{Mutation}$$

In our experiments $P = 15$. For every virus in the population, we evaluate its fitness given by $R_v(v_k^i, a)$. We then sample a new virus $\hat{v}^{i+1}$ based on the fitness values, in particular:

$$\mathbb{P}(\hat{v}^{i+1} = v_k^i) \propto \exp(\beta R_{\text{v}}(v_k^i, a))$$

With duplicates in the population being considered distinct, so that the likelihood of a particular variant increases with the number of duplicates. Furthermore, $\beta$ is a constant which reflects how random the selection process is, with $\beta \to \infty$ reflecting deterministic max-fitness selection. After $H$ generations, a full escape trajectory $\hat{\boldsymbol{v}} = [\hat{v}^0, \hat{v}^1, \hat{v}^2, \ldots, \hat{v}^H]$ has been generated and the simulated viral escape process ends.

### 4.3. Antibody Optimisation

We define the antibody fitness $F_{\hat{v}}^H(a)$ such that it represents the *true* objective of the antibody, which accounts for the viral escape. Given a horizon $H$ and starting virus $\hat{v}$, the antibody fitness is:

$$F_{\hat{v}}^H(a) = \mathbb{E}_{\hat{\boldsymbol{v}} \sim \text{Ev}(\hat{v}, a)} \left[ \frac{1}{H+1} \sum_{i=0}^{H} R_a(\hat{v}^i, a) \right] \quad (2)$$

Note that if $H = 0$ this fitness defaults to a naive antibody payoff that ignores viral escape, i.e., $F_{\hat{v}}^0(a) = R_a(\hat{v}, a)$. We refer to this as the *myopic* objective.

To optimise both shapers and myopic antibodies, we employ Monte Carlo simulations to estimate the antibody fitness, combined with an evolutionary optimisation algorithm. We refer to this process as the *antibody optimisation loop*, see Figure 2a and Algorithm 2. In meta-learning terms, this is the *outer loop* or the *meta-loop*, contrasting to the *inner loop*, which is the *simulated viral escape via evolution*.

Given a starting antibody $\hat{a}$, a starting virus $\hat{v}$ and a viral escape horizon $H$, the antibody optimisation process generates a trajectory of antibodies $\hat{\boldsymbol{a}} = [\hat{a}^0, \hat{a}^1, \hat{a}^2, \ldots, \hat{a}^N]$, where $N$ is the number of antibody optimisation steps (i.e., meta-steps). In the trajectory, $\hat{a}^0 = \hat{a}$ is the starting antibody and $\hat{a}_N$ is the final optimised antibody. This optimisation

could, in principle, start from any antibody, but for simplicity we opt to start from purely random antibodies, meaning $\hat{a}$ is random. In most of our experiments $N = 30$.

At the start of antibody optimisation step $i$, we have an antibody $\hat{a}^i$. We first generate a population of $P_a$ antibodies $[a_1^i, a_2^i \ldots a_{P_a}^i]$ by taking both the antibody $\hat{a}^i$ and $P_a - 1$ copies of it, with each copy having exactly a single random mutation in the amino acid sequence of $\hat{a}^i$. For our experiments, $P_a = 40$.

We then sample their fitness values $F_{\hat{v}}^H(a_j^i)$ with a fixed number $\eta$ of Monte Carlo roll-outs, i.e., we sample $\eta$ independent viral escape trajectories, each with horizon $H$ viral escape steps. We found $\eta = 5$ to be sufficient. Finally, we select $\hat{a}^{i+1}$ to be the best-performing antibody:

$$\hat{a}^{i+1} = \arg \max_k \mathbb{E} \left[ F_{\hat{v}}^H(a_k^i) \right]$$

Once the final optimised antibody $\hat{a}^N$ is generated, a full optimisation trajectory is complete, $\hat{\boldsymbol{a}} = [\hat{a}^0, \hat{a}^1, \hat{a}^2, \ldots, \hat{a}^N]$, and the antibody optimisation process finishes.

## 5. Experimental Setup

### 5.1. Absolut! Speedup

To meet the computational demands of our opponent shaping approach, which requires rapid evaluation of numerous antibody-virus interactions, we reimplement the core binding calculation of Absolut! (Robert et al., 2022) using JAX (Bradbury et al., 2018), a framework that facilitates GPU-accelerated computation (Figure 2c). Our efficient JAX implementation and the GPU acceleration results in a 10,000-fold speedup compared to the original implementation, see Table 1.

|  | Absolut! | Absolut! + JAX |
|---|---|---|
| Hardware | Apple M2 Max | Nvidia A40 |
| Time/Antigen (s) | 1.8 | $\mathbf{2.1 \times 10^{-4}}$ |

*Table 1.* Comparison of the time taken to compute a single binding query between the original implementation of Absolut! (Robert et al., 2022) and our reimplementation of Absolut! in JAX (Bradbury et al., 2018). The original Absolut! implementation runs on CPU only, hence the difference in evaluation hardware.

### 5.2. Dengue Virus

We use the antigen protein from the Dengue Virus for our main experiments, specifically, the structure with Protein Data Bank (PDB) code 2R29 (Berman et al., 2000; Lok et al., 2008). First, Absolut! processes this structure to generate binding-relevant information, which is then used by our JAX implementation (details given in Appendix D). In the viral escape step, we mutate only the amino acid sequence of

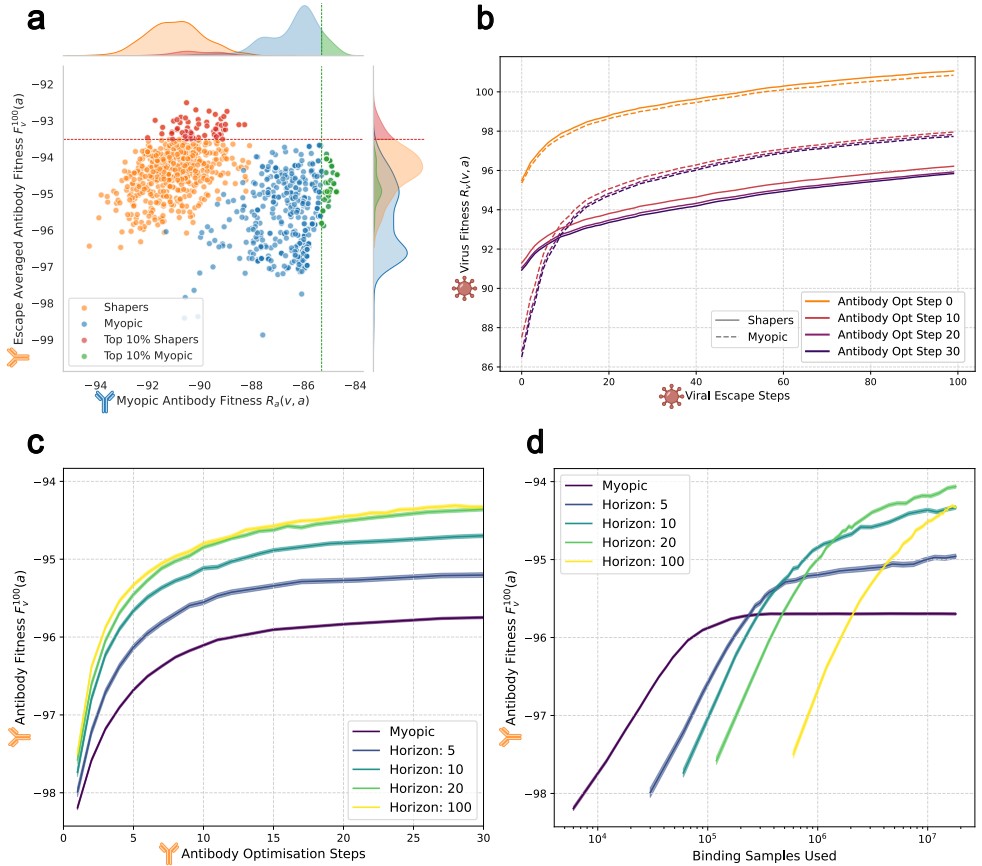

*Figure 3.* **Shapers Outperform Myopic Antibodies. a** Distribution of antibody shapers optimised with horizon $H = 100$ (orange) vs. myopic antibodies distribution (blue). We highlight the top 10% shapers with respect to $F_v^{100}(a)$ in red, and the top 10% myopic antibodies with respect to $R_a(v, a)$ in green. The x-axis is the myopic antibody fitness $R_a(v, a)$ and the y-axis is the escape averaged antibody fitness for $H = 100$, i.e., $F_{\hat{v}}^{100}(a)$. Higher values on both axes indicate better performance. **b** Viral escape curves (inner loop performance) for different steps of the antibody optimisation process (outer loop) for antibody shapers optimised with horizon 100 (solid lines) and myopic antibodies (dashed lines). The lighter lines indicate early antibody optimisation steps, and the darker lines show the later steps. The x-axis shows the evolutionary steps of viral escape. The y-axis represents the virus fitness/payoff $R_v(v, a)$, where higher values indicate better virus fitness (and lower values denote better antibody performance, so lower is better for us). **c, d** Antibody optimisation learning curves (outer loop performance) for a varying horizon length. The x-axis shows the antibody optimisation steps, i.e., meta-steps (**c**) or the number of samples from the binding simulator (**d**). The y-axis shows antibody fitness $F_v^{100}(a)$. Error bars correspond to the standard error. Higher values indicate better performance.

the dengue envelope antigen, which is composed of $N_v = 97$ amino acids, and do not consider other components of Dengue Virus. Importantly, we assume that the structure of the antigen does not significantly change over the course of viral escape. All experiments but the ones we discuss in Section 6.3 and Appendix A use the dengue virus.

### 5.3. Additional Viruses and Bacterium

To demonstrate robustness of the experimental results we achieve with ADIOS on dengue virus we conduct additional experiments with three other viruses and one bacterium. The three viral antigens we use are: West Nile Virus, PDB code 1ZTX (Nybakken et al., 2005); Influenza Neuraminidase Virus, PDB code 4QNP (Wan et al., 2015) and

MERS-CoV Virus with PDB code 5DO2 (Li et al., 2015). Furthermore, we show that ADIOS can be easily applied to other pathogens, such as bacteria, too. We perform an extra experiment with the Clostridium Difficile Bacterium, PDB code 4NP4 (Orth et al., 2014).

## 6. Results

### 6.1. Shapers vs. Myopic Antibodies

We validate the effectiveness of the antibody shapers in optimising the escape-averaged antibody fitness function $F_v^H(a)$ compared to myopic antibodies that only respond to the current virus $v$. For our shaper antibodies, we select a long horizon of $H = 100$ to capture extended viral

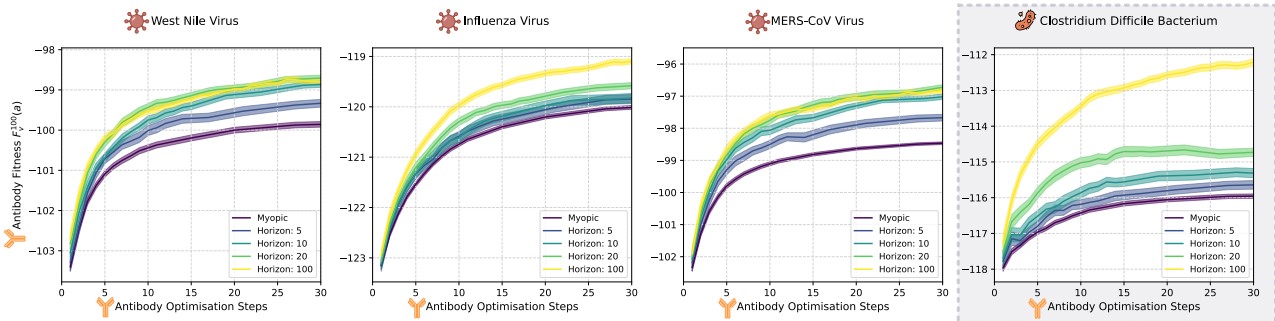

*Figure 4.* **Shapers Outperform Myopic Antibodies on Other Viruses and a Bacterium.** Antibody optimisation learning curves (outer loop performance) for varying horizon lengths across three viruses - West Nile Virus, Influenza Virus, and MERS-CoV - as well as the bacterium Clostridium Difficile. The x-axis shows the antibody optimisation steps, i.e., meta-steps and the y-axis shows the antibody fitness $F_v^{100}(a)$. Raw fitness values are dependent on Absolut! scores and are relative to the specific antigen, i.e., absolute values should not be compared between different viruses or bacterium but rather the overall trends. Error bars correspond to the standard error. Higher values indicate better performance. Full set of ADIOS results for these four pathogens is provided in Appendix A.

escape trajectories. Both shapers and myopic antibodies are optimised for $N = 30$ steps. Figure 3a presents the performance distributions of shapers and myopic antibodies under both objective functions.

Our results demonstrate a clear advantage of shapers in the escape-averaged objective $F_v^{100}(a)$. The mean of the shapers distribution significantly exceeds that of the myopic distribution, as evident from the marginal density plot in Figure 3a. Notably, none of the myopic antibodies outperform *any* of the top 10% of shapers in this long-term objective. However, there is a trade-off between short-term and long-term optimisation. While shapers do better on the escape-averaged objective, they underperform on the myopic objective $R_a(v, a)$.

We next examine the influence of antibody shapers on viral escape trajectories, comparing $H = 100$ shapers with myopic antibodies, both optimised for $N = 30$ steps. Figure 3b illustrates the viral escape curves induced by both antibody types at different stages of their optimisation process. We first complete the antibody optimisation process, saving antibodies generated at steps $0, 10, 20$, and $30$. For each of these optimisation steps, we then simulate viral escape over $H = 100$ evolutionary steps using the corresponding saved antibodies. The presented viral escape curves are averages derived from multiple simulations.

At the outset of the antibody optimisation process (step 0), both the shapers and the myopic antibodies induce similar escape curves, an expected outcome given their initialisation from random antibody sequences. However, as we examine antibodies from later optimisation steps, we observe diverging trends. Myopic antibodies cause the viral fitness to be lower in the initial escape steps, outperforming the shapers. After about 10 escape steps, corresponding to $\approx 10$ viral mutations, the two antibody types perform similarly. Beyond that, shapers demonstrate superior results in later escape stages, more effectively preventing viral escape.

These results show that as the antibody optimisation process progresses, shapers learn to influence viral trajectories in a way that minimises long-term viral escape, albeit at the cost of initial performance. While myopic antibodies may offer better immediate control, shapers provide more sustained effectiveness against evolving viral populations.

### 6.2. Antibody Shapers with Varying Horizons

Finally, we investigate the impact of varying horizons $H$ on the optimisation process of antibody shapers. We optimise myopic antibodies and shapers using horizons $H = \{5, 10, 20, 100\}$ for $N = 30$ steps. To evaluate these antibodies against a consistent "true" objective, we simulate viral escape over $H = 100$ steps for each antibody, regardless of the horizon used during its optimisation. Figure 3c presents these results, demonstrating that shapers optimised with longer horizons $H$ consistently yield better performance throughout all steps of the optimisation process.

However, the number of antibody optimisation steps does not accurately reflect the computational or experimental cost of optimisation. Each simulation of viral escape requires a number of binding samples that increases linearly with the horizon length $H$. Yet, shorter horizon antibodies optimise an objective that diverges further from our "true" antibody objective $F_v^{100}(a)$. Due to this trade-off, we observe that the optimal training horizon varies depending on the available computational budget.

To illustrate this trade-off, we conduct an additional experiment shown in Figure 3d. Here, instead of fixing the number of optimisation steps $N$, we constrain the total number of binding samples - queries to our binding strength simulator used to evaluate all antibody and virus payoffs throughout the optimisation process - to be constant across different horizons. This approach provides a performance comparison that accounts for the computational resources

necessary across varying horizon lengths. Interestingly, $H = 20$ shapers perform strongly, nearly matching the performance of those optimised with horizon $H = 100$ for a given number of antibody optimisation steps, and far exceeding it when accounting for the differing computational cost. This suggests that using a cheaper, shorter-horizon proxy for the true antibody objective $F_v^{100}(a)$ can yield substantial benefits.

More generally, we find that the optimisation horizon significantly influences the performance of antibody shapers. While longer horizons lead to better long-term performance, the optimal horizon length is dependent on the available computational resources. Thus, it is important to consider the balance between computational cost and the fidelity of the optimisation objective when designing antibodies for long-term effectiveness against evolving viral populations.

### 6.3. Antibody Shapers for Other Viruses and Bacterium

To test whether the shaping effects observed on dengue generalise, we evaluate ADIOS on three additional viruses: West Nile, Influenza, and MERS-CoV; as well as the Clostridium Difficile bacterium (Figure 4, Appendix A). In all cases, we observe consistent trends: shapers outperform myopic antibodies in escape-averaged fitness, confirming that ADIOS can successfully achieve shaping across diverse pathogens.

For West Nile virus and MERS-CoV, $H = 100$ shapers appear to perform worse than shorter-horizon $H = 20$ antibodies, see Figure 4. However, given that all antibodies are only optimised for 30 meta-steps, and the compute-normalised (bottom row) plots in Figure A.1 show clearly that $H = 100$ shapers have not yet converged, we hypothesise that $H = 100$ shapers would ultimately yield the best performance if given more optimisation time.

Interestingly, the shaping effect is especially strong on the Clostridium difficile bacterium, where $H = 100$ shapers significantly outperform all other antibody types across all reported metrics; see right-most row in Figure A.1. These results suggests that different antigens can exhibit very different behaviour in a shaping setting. However, ADIOS is able to produce antibodies with effective shaping behaviour across them.

### 6.4. Attack is the Best Defence

Our previous results demonstrate that antibody shapers, particularly those optimised with longer horizons, manage to effectively minimise viral escape. However, we hypothesise they can achieve this through two distinct strategies: *robustness* or *shaping*. A robustness strategy involves developing antibodies that are inherently resistant to a wide range of potential viral variants — a "good defence" approach. In contrast, a shaping strategy aims to actively influence the evolutionary trajectory of the virus itself, creating evolutionary pressures that guide viral mutations in a direction more favourable to antibody binding — an "attack" approach.

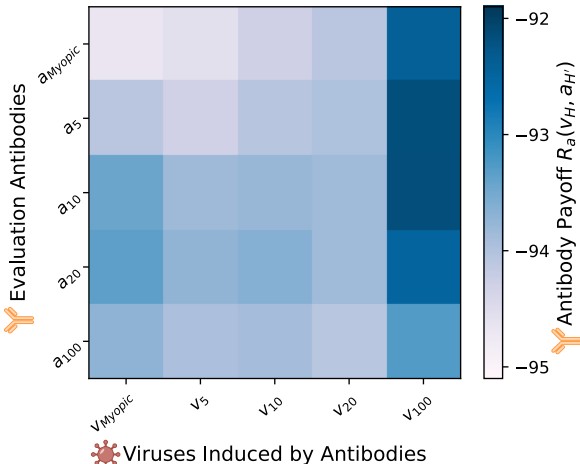

*Figure 5.* **Robustness vs. Shaping.** We optimise 80 different antibodies $a_H$ across multiple horizons (*Myopic*, $H = 5, H = 10, H = 20, H = 100$), these are represented by the y-axis. We simulate the viral escape to each of these antibodies for 100 steps, and we group the escape viruses $v_H$ by the horizon $H$ of the antibody that induced them; these escape viruses are represented by the x-axis. In colour, we show the mean antibody payoff $R_a(v_H, a_{H'})$ for each group of optimised antibodies $a_{H'}$ against the final escape viral variant $v_H$ induced by other antibodies optimised with horizon $H$. Darker colours correspond to better antibody payoff.

To disentangle these strategies, we *separately* evaluate the antibodies and the viruses that evolve in response to them. To do this, we compare the viruses against *other* antibodies which *did not* influence the viral evolution. The intuition is that an antibody which is good at shaping (good attack), but less robust (poor defence), will induce viruses which *other* antibodies will perform well against. Specifically, we generate antibodies $a_H$ for each horizon $H$ and simulate viral escape against these antibodies for 100 steps, resulting in viruses $v_H$. For all pairs of horizons $(H, H')$, we then cross-evaluate the antibody payoff $R_a(v_H, a_{H'})$. Figure 5 presents the result of this analysis.

Interestingly, viruses $v_{100}$ induced by $H = 100$ shapers are consistently more exploitable by antibodies across all optimisation horizons. This suggests that $H = 100$ shapers actively shape the escape trajectories of the virus in a way that makes the resulting variants more susceptible to antibody binding in general. However, this shaping effect comes at a cost. The $H = 100$ shapers ($a_{100}$) show slightly lower payoffs compared to the peak performance of shorter-horizon antibodies ($a_5$ and $a_{10}$) against the viruses $v_{100}$ induced by the $H = 100$ shapers (see rightmost column of Figure 5). This trade-off indicates that to exert a stronger shaping influence on viral evolution, $H = 100$ shapers sac-

rifice some degree of immediate performance or robustness, that is, their ability to perform well against a wide range of viruses. Therefore, a potential strategy could involve using a mixture of antibodies as therapy, where some are optimised for shaping the virus's evolutionary trajectory, and others are designed for strong immediate binding.

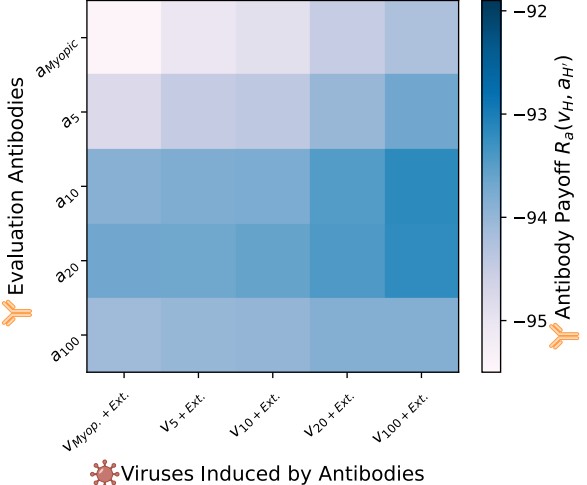

*Figure 6.* **Shaping with External Pressure.** A similar experiment to Figure 5, but with the additional external pressure of a separate myopic antibody (see Equation 3).

To investigate whether shaping persists in more realistic scenarios with multiple therapeutic pressures, we conduct an additional experiment. Using the same groups of antibodies $a_H$, we simulate viral escape with external pressure from a myopic antibody $a_{Ext}$ (not included in the original $a_{myopic}$ set). The viruses $v_{H+Ext}$ now evolve according to a modified payoff:

$$R_v^{Ext}(v, a, a_{Ext}) = \frac{1}{2}R_v(v, a) + \frac{1}{2}R_v(v, a_{Ext}) \quad (3)$$

which represents the scenario where multiple therapies are present in the environment (for example, during COVID-19 when multiple vaccines were available). Figure 6 shows that while the shaping effect is somewhat reduced compared to our original results, it remains clearly visible. This demonstrates that our shaping approach transfers to test regimes where external pressures from other therapies are present.

### 6.5. Explainability Analysis

To understand what distinguishes shapers from myopic antibodies, we conduct two complementary analyses of amino acid distributions and binding poses (Appendices B and C). Examining amino acid distributions, we find that long-horizon shapers exhibit more uniform distributions, while myopic antibodies tend to cluster around amino acids with extreme binding energies. We hypothesise that by maintaining diversity in their amino acid composition, shapers can

preserve robustness against viral mutation since the virus cannot easily escape by avoiding specific, high-binding, parts of the antibody.

Through analysis of binding using pose matrices, representing which parts of the antibody and virus bind with each other, we observe these interaction patterns significantly change as the virus adapts. However, the type of antibody influences the nature of these changes: $H = 100$ shapers actively constrain viral evolution by both preventing unfavourable binding configurations and preserving favourable ones. While these findings are specific to our Absolut! binding simulator, they hint at explainable strategies that shapers use to influence viral evolution, which could inform future antibody design approaches.

## 7. Conclusion & Future Work

In this work, we introduce ADIOS, a meta-learning framework for designing therapeutic antibodies that not only defend against current viral strains but instead *actively shape viral evolution*. We provide a GPU-accelerated JAX implementation of Absolut!, enabling rapid simulation of viral escape trajectories and outer-loop optimisation. Our results demonstrate that shapers are not only more robust against viral escape, but they also shape viral evolution toward more targetable variants. Lastly, we provide an explainability analysis of how shapers achieve this level of robustness and influence, which we hope will inspire practitioners.

Although dengue virus served as our primary benchmark, we evaluated ADIOS on three other viruses (West Nile, Influenza and MERS-CoV) and on the bacterium Clostridium Difficile. In all four cases ADIOS consistently achieves shaping. These results confirm that ADIOS can generalise across a diverse set of pathogens. More broadly, the same opponent-shaping principle can be transferred to monoclonal-antibody (mAb) therapy for cancer (Zahavi & Weiner, 2020). In that setting, the outer loop would optimise therapeutic mAbs, while the inner loop would simulate the evolution of cancer-cell growth-factor receptors; the goal would be to shape cancer cells into cells that do not proliferate well. Exploring this direction, together with other bacterial or antimicrobial-resistance scenarios, remains an exciting avenue for future work.

While our current implementation uses simplified binding and evolutionary escape models that prevent direct therapeutic application, ADIOS could be integrated with more sophisticated models, like AlphaFold3 (Abramson et al., 2024), to better capture evolving viral and antibody structures. As computational models of protein interactions and evolutionary processes continue to improve, ADIOS has the potential to transform how we develop therapies against viruses, cancers, and other evolving adversaries.

## Acknowledgements

We would like to thank Marius Urbonas, Duygu Açıkalın, Michael Matthews, Benjamin Ellis, Benedetta L. Mussati, Mattie Fellows, Lisa Zillig, Ting Lee, Matthew Raybould and Charlotte Deane for their invaluable contributions to this work. Their thoughtful insights helped guide the project direction, their detailed feedback on earlier drafts significantly enhanced the manuscript's clarity and accessibility, and their support during the review process was instrumental in addressing reviewers' concerns. We also thank the anonymous reviewers for their constructive feedback, which led to improvements in the paper. This work was supported by UK Research and Innovation and the European Research Council - Jakob Foerster is partially funded by the UKRI grant EP/Y028481/1, originally selected for funding by the ERC. This work was also supported by Exscientia, which provided Aleksandra Kalisz with a PhD studentship.

## Impact Statement

This work introduces a machine learning framework for antibody design that could contribute to developing more effective therapies against evolving pathogens like viruses. While this has a potential positive societal impact through improved disease treatment and pandemic preparedness, we acknowledge that therapeutic development tools carry risks if misused. Our current implementation uses simplified models and significant additional research and safety validation would be required before any real-world therapeutic applications.

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

## A. Experimental Results on Other Viruses and Bacterium

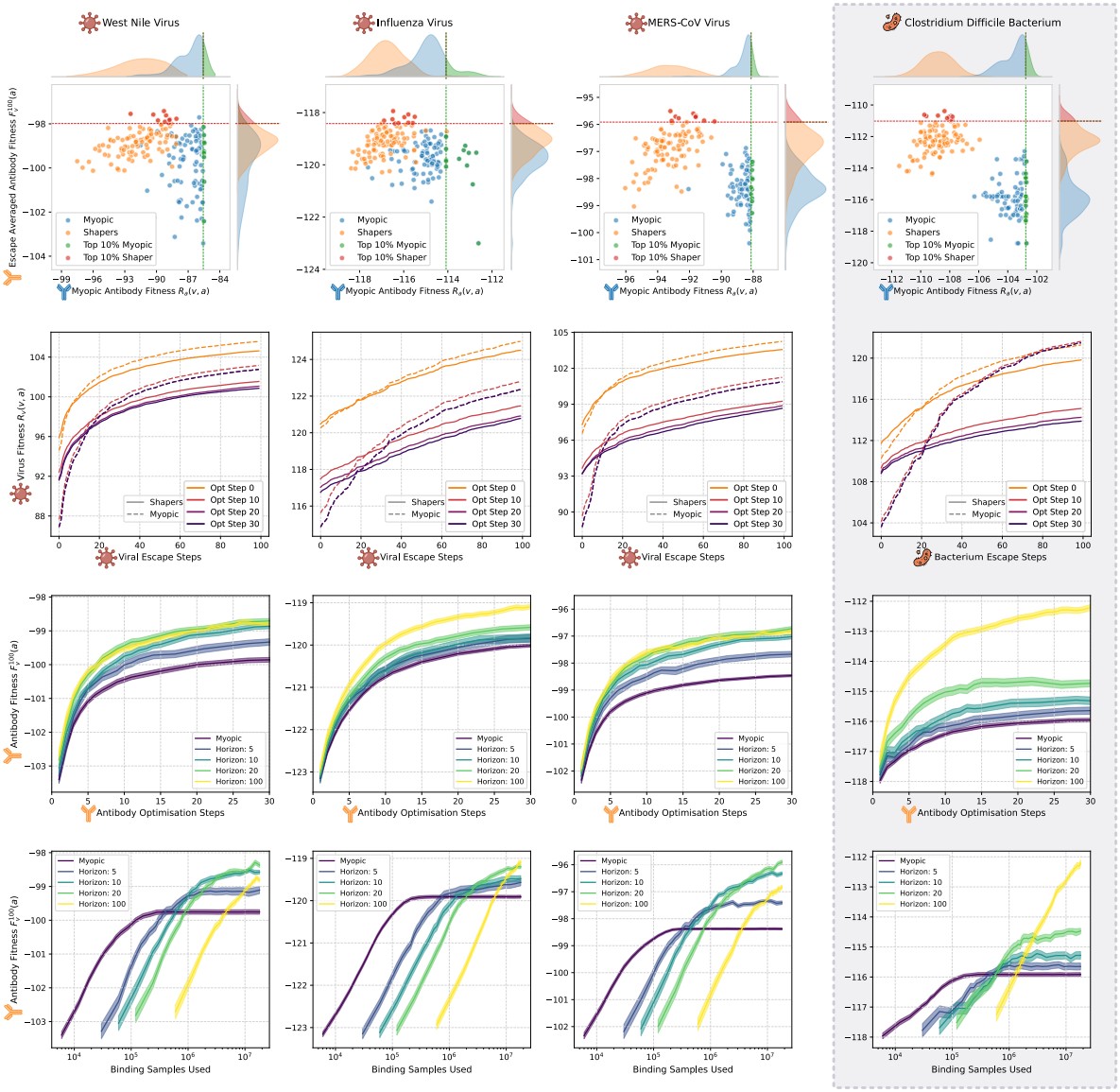

*Figure A.1.* **Shapers Outperform Myopic Antibodies on Other Viruses and a Bacterium.** The figure shows results across four different pathogens: West Nile Virus, Influenza Virus, MERS-CoV Virus, and Clostridium Difficile Bacterium (left to right columns). **First row** Distribution of antibody shapers optimised with horizon $H = 100$ (orange) vs. myopic antibodies distribution (blue). We highlight the top 10% shapers with respect to $F_v^{100}(a)$ in red, and the top 10% myopic antibodies with respect to $R_a(v, a)$ in green. The x-axis is the myopic antibody fitness $R_a(v, a)$ and the y-axis is the escape averaged antibody fitness for $H = 100$, i.e., $F_v^{100}(a)$. Higher values on both axes indicate better performance. **Second row** Viral escape curves (inner loop performance) for different steps of the antibody optimisation process (outer loop) for antibody shapers optimised with horizon $H = 100$ (solid lines) and myopic antibodies (dashed lines). The lighter lines indicate early antibody optimisation steps, and the darker lines show the later steps. The x-axis shows the evolutionary steps of viral escape. The y-axis represents the virus fitness/payoff $R_v(v, a)$, where higher values indicate better virus fitness (and lower values denote better antibody performance, so lower is better for us). **Third row** Antibody optimisation learning curves (outer loop performance) for varying horizon lengths. The x-axis shows the antibody optimisation steps, i.e., meta-steps, and the y-axis shows antibody fitness $F_v^{100}(a)$. **Forth row** Antibody optimisation learning curves accounting for computational cost. The x-axis shows the number of samples from the binding simulator, and the y-axis shows antibody fitness $F_v^{100}(a)$. Higher values indicate better performance. Error bars correspond to the standard error. In all these results, raw payoff/fitness values are dependent on Absolut! scores and are relative to the specific antigen, i.e., absolute values should not be compared between different viruses or bacterium but rather the overall trends.

## B. Amino Acid Distribution in Shapers and Myopic Antibodies

To further understand the performance differences between myopic antibodies and shapers, we analyse how the amino acid distributions of the antibodies change with the optimisation horizon $H$. Within our model, each amino acid is solely characterised by its binding strength to other amino acids as defined by the Miyazawa-Jernigan energy potential matrix (Miyazawa & Jernigan, 1999). However, despite this simplicity, we still see interesting patterns in the amino acid distribution. Figure B.1 showcases the results of our experiment.

Antibodies optimised with longer horizons, especially the $H = 100$ shapers, exhibit a more uniform distribution of amino acids, while those with shorter horizons show a tendency to cluster around amino acids associated with either high or low binding energies. The flatter distribution of long-horizon shapers suggests a more diverse and balanced approach to viral antigen binding. We hypothesise that this strategy helps to preserve robustness against viral mutations. By maintaining a more even distribution across energy levels, these antibodies may be less susceptible to viral escape.

In contrast, the clustering behaviour we observe in shorter-horizon antibodies indicates a more specialised strategy. By concentrating on amino acids at the extremes of the binding energy spectrum, these antibodies may achieve strong immediate binding but potentially at the cost of long-term robustness. However, while this analysis hints at the robustness of long-term shapers, it does not fully explain the shaping behaviour we observed in our previous results. In the next section, we investigate the distribution of amino acids within specific binding poses.

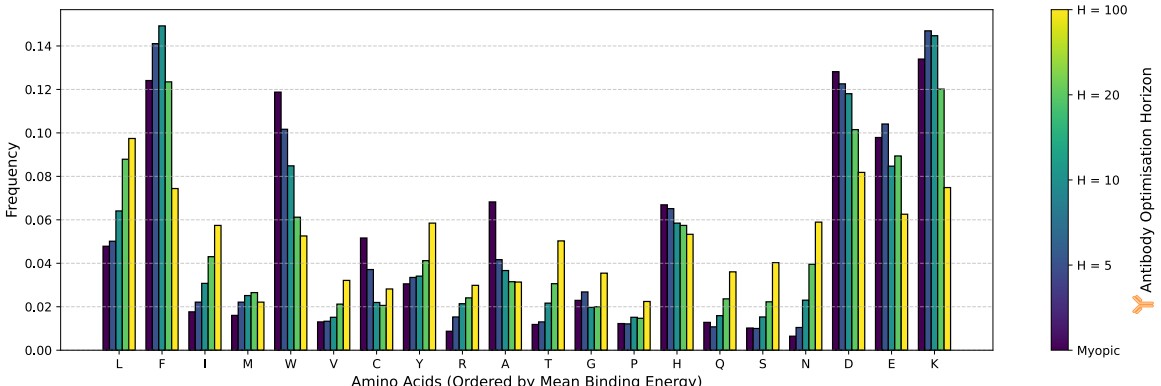

*Figure B.1.* **Distribution of amino acids in myopic antibodies and shapers.** The antibodies are optimised for $N = 30$ steps using different viral escape horizons $H$. Longer horizon shapers push the amino acid distribution closer to a uniform distribution.

## C. Influence of Antibody Shapers on Binding Poses

In the Absolut! framework (Robert et al., 2022), binding poses are defined as sets of interacting residue pairs between the antibody and the antigen. The binding energy of a pose is calculated by populating these residue locations with the amino acid sequences of both the antibody and the virus and then summing the pairwise interaction energies defined by (Miyazawa & Jernigan, 1999). Absolut! considers a vast number of possible poses (on the order of $10^6$) and determines the overall interaction energy as the energy of the minimum pose, refer to Appendix D for more details. Importantly, only a small part of the viral sequence contributes to this minimum energy pose.

As both the virus and the antibody mutate during our optimisation process, the lowest energy pose can change. To capture these dynamics, we introduced the concept of a pose matrix: a $20 \times 20$ matrix with one entry for each possible pair of amino acids. One dimension corresponds to the antibody amino acids, and the other to the viral amino acids. The entries in this matrix represent the number of interactions between the specific amino acid pairs in the lowest energy pose for the binding configuration between an antibody and an antigen.

Figure C.1a presents average pose matrices from multiple optimisation runs of both myopic antibodies and long-horizon shapers. We observe two key trends. First, as viral escape steps increase (top row vs bottom row), the pose matrices become more "diffused". This is expected, as the virus explores more "pose possibilities" through mutations during escape. Second, as the horizon of antibody optimisation increases, the poses also become more "diffused". This is particularly interesting, as all antibodies have the same number of mutations regardless of the horizon, suggesting that this diffusion might relate to the increased robustness of shapers.

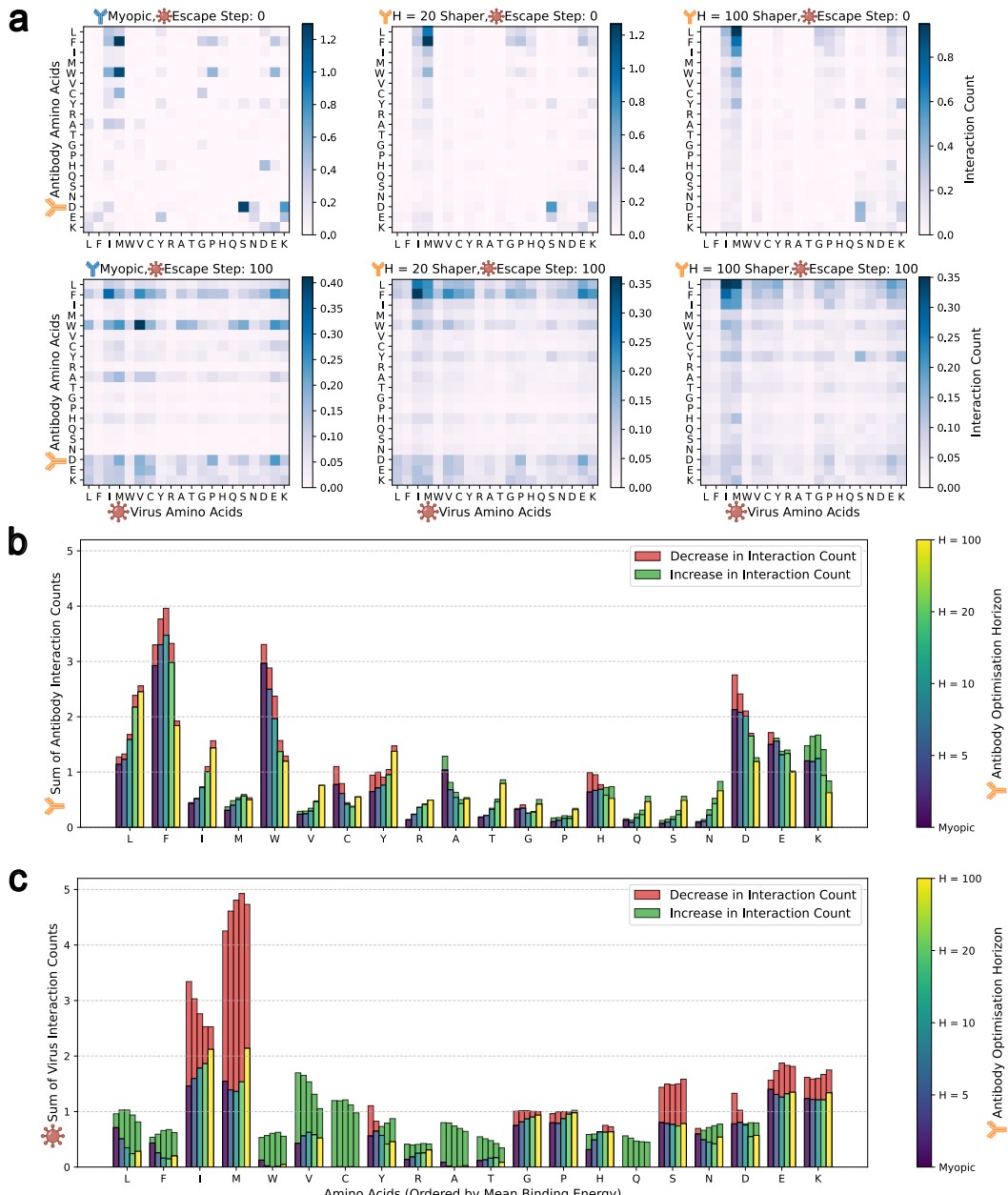

*Figure C.1.* **Influence of antibody shapers on binding poses. a** Average pose matrices between antibodies optimised using different horizons and the virus at various stages of its escape. The escape steps increase from left to right, and the horizon increases from top to bottom. The full grid of matrices with more antibody horizons and virus escape steps is available in the Supplementary Information, Figure C.2. **b, c** Aggregated sum of pose matrices w.r.t the antibody axis (**b**) and w.r.t the virus axis (**c**). The plots show a change in the interaction counts in the poses from the viral escape step 0 to 100. Red indicates a decrease in the interaction count and green an increase.

To further understand these pose dynamics, we aggregate the pose matrices along the antibody axis (Figure C.1b) and the virus axis (Figure C.1c). These figures show the change in interaction counts between viral escape steps 0 and 100. Figure C.1b shows that as the virus escapes it includes more of the antibody's lowest binding amino acids (particularly K, Lysine) in the pose. Notably, long-horizon shapers, especially $H = 100$ shapers, are most effective at preventing this increase in K (Lysine) interactions. Furthermore, Figure C.1c shows another viral escape strategy, where the virus removes its high-binding amino acids I (Isoleucine) and M (Methionine) from the pose. Again, $H = 100$ shapers are most successful in counteracting this trend, although they cannot completely prevent it.

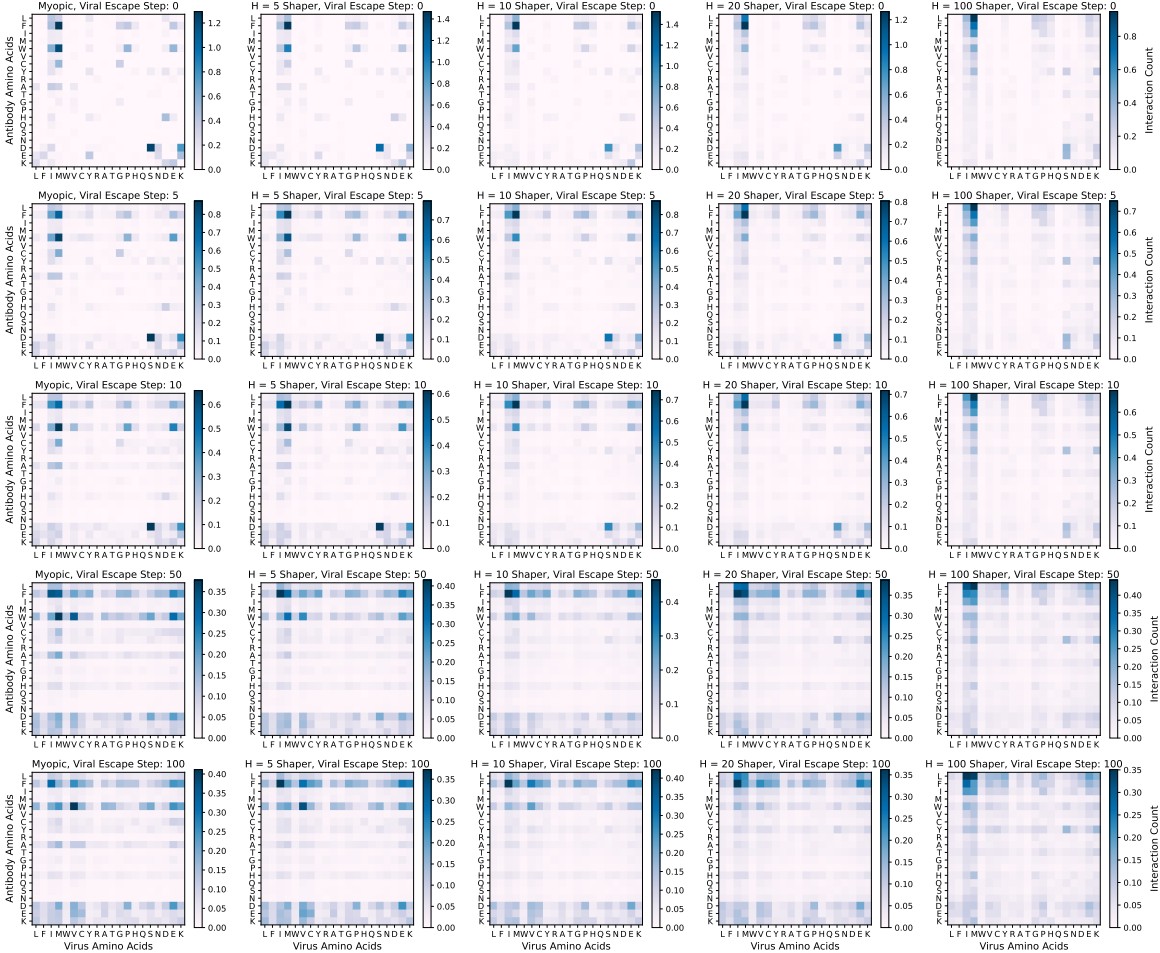

*Figure C.2.* Full grid of pose matrices. They represent the average pose interactions between myopic antibodies or antibody shapers optimised with different horizons and viruses at different stages of their escape.

Based on these observations, we hypothesise that the shaping ability of $H = 100$ shapers relies on two main mechanisms. Preventing the virus from including the antibody's lowest binding amino acids in the pose, and inhibiting the virus from removing its own high-binding amino acids from the pose. These strategies constrain the viral escape trajectories, making the resulting viral variants more susceptible to antibody binding in general. While these results are specific to our Absolut! binding simulator, they demonstrate that the behaviour of antibody shapers is both explainable and intuitive. This work serves as a proof of concept, showing that opponent shaping techniques can optimise antibodies to more effectively prevent viral escape.

## D. Binding Function

In general, ADIOS is independent of the choice of the binding function $B : \mathbb{A}^{N_v} \times \mathbb{A}^{N_a} \to \mathbb{R}$. In our work, we rely on the Absolut! framework (Robert et al., 2022) to implement the binding function. In this section, we mathematically formalise the binding energy calculation that Absolut! uses. For further explanation, readers are recommended to refer to the original Absolut! paper (Robert et al., 2022).

For two given protein structures, there are many possible joint configurations. Each of these joint configurations yields an energy. The configurations which are associated with lower energy will require more external energy to cause the system to leave that state, meaning in turn that they are more stable. If the configuration is sufficiently stable, this may be referred to as a binding pose.

In Absolut, poses are represented as pairs of residues[2] which are adjacent to each other in that pose. In particular, the pairs may be from the antigen to the antibody, or from the antibody to itself. We define the space of possible poses $\Phi$:

$$\Phi = 2^{N_v \times N_a} \times 2^{N_a \times N_a}$$

Where $N_v$ and $N_a$ are taken to be the set of integers up to $N_v$ and $N_a$, respectively.

The energy of a complex of a virus $v \in \mathbb{A}^{N_v}$ and an antibody $a \in \mathbb{A}^{N_a}$, in a given pose $(\phi^{v \times a}, \phi^{a \times a}) \in \Phi$ is defined by sum of the energies of each adjacent residues. The energy between a residue pair is determined by which two amino acids it contains, given by a symmetric *interaction matrix* $M : \mathbb{A} \times \mathbb{A} \to \mathbb{R}$, which is determined experimentally (Miyazawa & Jernigan, 1999).

We then define the energy of a single pose to be:

$$\hat{E}(a, b; (\phi^{v \times a}, \phi^{a \times a}), M) = \sum_{(i,j) \in \phi^{v \times a}} M(v_i, a_j) + \sum_{(i,j) \in \phi^{a \times a}} M(a_i, a_j)$$

Finally, given a set of poses $S \subseteq \Phi$, the binding strength is:

$$B(v, a) = -E(v, a; S, M) = -\min_{\phi \in S} \hat{E}(v, a; \phi, M) \tag{4}$$

Absolut! generates $S$ through a two-step process. First, Absolut! discretises a given structure of the virus $v$ (or any antigen) which is taken from the PDB (consortium, 2018). Second, Absolut! does a brute-force search over possible (discretised) poses for an antibody $a$ to join to the viral structure. The exact details are not necessary for this paper, we refer interested readers to the original paper.

However, we find that Absolut! generates more poses than we require. Since the energy function, $E$ is a minimum over poses, certain poses contribute far more than others. In particular, if a pose $\phi$ tends to yield higher energies, so $\hat{E}(a, b; \phi, M)$ is relatively large, it will have little impact on the result of $B$.

To give a more concrete example, for this paper, we use the dengue virus antigen (Lok et al., 2008). Absolut! gives $\approx 1.5$ million poses for this structure. Absolut! also comes with $\approx 20$ million real-world antibody sequences. When using the base dengue sequence as the antigen, across the 20 million binding calculations only 1027 binding poses are ever the minimum. Furthermore, the relevance of each pose drops exponentially. The most relevant pose accounts for $20\%$ of binding configurations, and by using the top 100 poses one would get the exact same result for binding in $95\%$ of antibodies. This gives us a way to make the computation 1000 times faster[3] for a negligible accuracy drop for this particular antigen sequence.

However, this leads to more errors as soon as we change the viral antigen sequence. Looking at the particular poses which lead to binding reveals another way to cut down on the total number of poses: all of the poses contain at least 18 pairs of residues. As the interaction matrix $M$ is strictly negative, having more pairs of residues always makes the binding energy of a pose lower, meaning it is more likely to be where binding occurs. Out of the original 1.5 million poses, only approximately 37 thousand (1 in 40) contain 18 or more pairs of residues. When using only these residues, we see no differences across any of the evolutionary simulations. It is possible that a pose with 17 or less pairs is the dominant one for some antigen $v$ with antibody $a$, but if so, then such pairs appear to be extremely rare.

Using these methods of pruning poses gives us two subsets of the original set of poses, a larger one which almost exactly matches performance, and another which sometimes differs, but is much faster to compute. We refer to these as the *high-resolution* and *low-resolution* binding simulators, respectively. Note that for the low-resolution binding simulator, the more mutations the virus undergoes, the less accurate it becomes. Furthermore, we also do binding to the antibody anti-target, $t_a^-$. To account for this, we compute the relevant poses for this anti-target too.

When running dengue virus experiments, we always train with the low-resolution binding simulator, then perform "verification" with the high-resolution one, and these are the results we report throughout the paper. For the other viruses and the bacterium we run "verification" in a differently initialised instantiation of the low-resolution simulator. The reason is

---

[2]A single amino acid position on a protein

[3]In practice, the difference is closer to $10,000$, likely due to the GPU having to move less data.

twofold. Firstly this enables us to run many more evolution experiments. Secondly, this mimics the real-life process of transferring out of simulation to the real world. By showing we transfer from the low-resolution binding simulation to the slower, high-resolution binding simulation, we demonstrate that our results are not extremely specific to the exact simulation we use and that any result will not disappear as soon as a more accurate simulation is used. We emphasise that Absolut! does not represent an accurate model of antibody binding. It is instead a toy simulation to demonstrate our methodology. For example, we do not expect our framework *when used with this simulation model* to yield highly effective, superior antibodies for real-world applications.

## E. MDP for the Virus–Antibody Game

We formalise a *single interaction round* between an antibody and a virus as the finite–horizon Markov Decision Process $\mathcal{M}$:

$$\mathcal{M} = \langle \mathcal{S}, \mathcal{A}^{\mathrm{v}}, \mathcal{A}^{\mathrm{a}}, P, R, \mu \rangle.$$

- **State space**: $\mathcal{S} = \mathbb{A}^{N_v} \times \mathbb{A}^{N_a}$, where a state $s = (v, a)$ is the pair of viral ($v$) and antibody ($a$) amino-acid sequences.

- **Action spaces** (chosen simultaneously):

$$\mathcal{A}^{\mathrm{v}}(s) = \mathbb{A}^{N_v}, \qquad \mathcal{A}^{\mathrm{a}}(s) = \mathbb{A}^{N_a}.$$

- **Transition kernel** (episode terminates after one step):

$$P\big(s' \mid s, a^{\mathrm{v}}, a^{\mathrm{a}}\big) = \mathbf{1}_{s'=s_\star}, \quad s_\star \text{ is terminal.}$$

- **Reward vector** $R = (R_{\mathrm{v}}, R_{\mathrm{a}})$. Given joint action $\big(a^{\mathrm{v}}, a^{\mathrm{a}}\big)$ in state $s$,

$$R_{\mathrm{a}}\big(s, a^{\mathrm{v}}, a^{\mathrm{a}}\big) = B\big(a^{\mathrm{v}}, a^{\mathrm{a}}\big) - B\big(t_a^-, a^{\mathrm{a}}\big) - B\big(a^{\mathrm{v}}, t_v^+\big), \qquad R_{\mathrm{v}} = -R_{\mathrm{a}}.$$

- **Initial-state distribution** $\mu$ puts mass on the wild-type virus and the initial antibody candidate: $\mu\big(v_0, a_0\big) = 1$.

Because this is a single-step MDP, the return equals the immediate reward $R$ and the discount factor is irrelevant.

