# OpenReview forum: "ADIOS: Antibody Development via Opponent Shaping"
_ICML.cc/2025/Conference — ICML 2025 poster_

### Official Review · Reviewer_kBGh · 2025-03-13

**Overall Recommendation:** 3

**Summary:**

The paper introduces ADIOS (Antibody Development via Opponent Shaping), a meta-learning framework that designs antibodies capable of both neutralizing current viral strains and influencing viral evolution to favor less dangerous variants. By framing antibody-virus interactions as a two-player zero-sum game, the method uses nested optimization loops: an inner loop simulating viral escape and an outer loop optimizing antibodies against long-term viral adaptation. ADIOS is implemented within the Absolut! framework, leveraging GPU acceleration for a 10,000x speedup. The authors demonstrate that ADIOS-optimized antibodies, or "shapers," outperform conventional myopic antibodies in long-term efficacy, shaping viral evolution to produce more targetable variants.

**Claims And Evidence:**

- The paper claims that ADIOS-optimized antibodies shape viral evolution toward weaker, more targetable variants. This claim is supported through simulation results demonstrating that shapers lead to viruses more susceptible to a broader range of antibodies.
- The claim that ADIOS outperforms myopic antibodies in long-term protection is substantiated by comparative evaluations, showing superior performance over extended evolutionary trajectories.
- The authors suggest that ADIOS has applications beyond antiviral therapy, including antimicrobial resistance and cancer treatment. While plausible, this claim remains speculative and would require further validation in these domains.
- The paper asserts that its GPU-accelerated JAX implementation of Absolut! achieves a 10,000x speedup. The performance comparison is well-documented, demonstrating substantial computational gains.

**Essential References Not Discussed:**

- The paper sufficiently engages with prior work on opponent shaping, reinforcement learning, and antibody-virus interactions.

**Experimental Designs Or Analyses:**

- The experimental setup effectively evaluates the model’s ability to shape viral evolution and sustain long-term efficacy.
- The choice of the dengue virus as a test case is reasonable, though additional validation on other viruses would strengthen the study.
- The paper provides clear analyses of the computational trade-offs in shaping horizons, offering practical guidance for deploying ADIOS in resource-constrained settings.

**Methods And Evaluation Criteria:**

- The method is evaluated within the Absolut! simulation framework, which is appropriate for modelling antibody-virus interactions.
- The evaluation compares ADIOS to myopic antibodies, demonstrating superior long-term efficacy. However, additional comparisons against existing evolutionary models would further validate ADIOS’s impact.
- The paper explores the trade-offs between short-term efficacy and long-term viral shaping, providing valuable insights into practical applications.

**Other Comments Or Suggestions:**

NA

**Other Strengths And Weaknesses:**

- Strengths:
  - ADIOS introduces a novel approach to antibody design by explicitly modeling viral adaptation, moving beyond conventional myopic strategies.
  - The study provides empirical validation, demonstrating that shapers outperform myopic antibodies in long-term efficacy.
  - The GPU-accelerated JAX implementation significantly improves computational efficiency, making large-scale evolutionary simulations feasible.
  - The exploration of shaping trade-offs provides valuable insights for real-world deployment.
  - The method has potential applications beyond virology, including antimicrobial resistance and cancer treatment.

- Weaknesses:
  - While the study presents compelling results, additional validation on a broader range of viruses would strengthen its generalizability.
  - The claim that ADIOS is applicable to antimicrobial resistance and cancer therapy remains speculative without experimental evidence.
  - The evaluation focuses primarily on simulation results, and real-world validation would be necessary to confirm ADIOS’s practical impact.
  - The authors assume that the structure of the antigen does not significantly change over the course of viral escape.

**Questions For Authors:**

- Have you considered evaluating ADIOS on other viruses beyond dengue to test its generalizability?
- How sensitive is ADIOS to different choices of evolutionary parameters in the viral escape simulation?
- How would you incorporate dynamics (going beyond the assumption of static antigen structure), to improve accuracy?
- Although it might be difficult, how would go about real-world validation?
- How do you envision ADIOS being applied to cancer therapy, and what adaptations would be required for such use cases?

**Relation To Broader Scientific Literature:**

- The work builds on prior research in computational antibody design, reinforcement learning, and viral escape modeling.
- The integration of opponent shaping into antibody therapy design represents a good contribution that extends beyond traditional machine learning-based antibody optimisation.
- The study effectively situates ADIOS within the broader context of viral adaptation and therapy resistance, demonstrating its relevance to long-term immunotherapy strategies.

**Theoretical Claims:**

- The theoretical framing of ADIOS as a two-player zero-sum game is well-grounded in reinforcement learning and opponent shaping literature.
- The meta-learning approach to optimising antibodies across viral evolutionary trajectories is effectively justified.

---

> ### Author Rebuttal · Authors · 2025-03-31
>
> Thank you very much for your thorough review and feedback! We appreciate it a lot. We are glad that you thought that applying opponent shaping to antibody design is "a good contribution that extends beyond traditional ML-based antibody optimisation"!
>
> # Answering Questions
> ## 1. Evaluating ADIOS on Other Viruses
> We agree that testing ADIOS against a wider range of viruses would help further validate our claims. To address this, we have conducted additional shaping experiments with three more viral antigens: the flu, MERS, and the West Nile virus. For all three, our experiments show that the antibody shapers generated by ADIOS successfully shape the new viruses, limiting viral escape. Interestingly, the computational trade-offs of shaping horizons are dependent on the virus but follow the same main trends. We have added these new results to the paper.
> ## 2. Sensitivity of ADIOS to Hyperparameters
> Based on our experiments ADIOS is not very sensitive to choices of different evolutionary parameters. As an example, we experimented with different mutation rates of the virus and found qualitatively similar results. To test the transferability of our results we also designed the external pressure experiment, the results of the experiment are in Figure 4. This result demonstrates the robustness of ADIOS to a domain shift. Even with an added pressure of an external antibody we can still see a shaping effect induced on the virus by the long horizon shapers. Additionally, the new results of the extra experiments on 3 other viruses show that ADIOS achieves shaping and limits viral escape for all the selected viruses, not only Dengue. Overall, this demonstrates that ADIOS is not sensitive to the choice of evolutionary parameters.
> ## 3. Incorporating Dynamics
> Absolut! already allows us to incorporate an aspect of ‘dynamics’, as it tests binding over a larger number of potential poses, and the antibody structure is permitted to change between poses. The minimum energy poses change dramatically throughout the evolution of antibody/antigen, and we study the influence of shapers on binding poses in Appendix B.
>
> That said, you are completely correct about the static structure of the antigen which is used to generate these binding poses in Absolut!. In future work, we plan to directly account for the changes in the antigen structure by using structure prediction models like AlphaFold3 (AF3), which can now accurately predict the structure of an antibody-antigen complex. We are building a binding simulator that combines the AF3 structure with a protein-protein docking score to achieve a more accurate binding affinity score. We plan to use this new binding simulator which accounts for antigen structure changes, instead of Absolut!, to further evaluate the ADIOS framework and demonstrate its real-world applicability.
> ## 4. Real-World Validation
> We want to conduct a real-world evaluation of ADIOS in the new simulator mentioned above by using data from past COVID variants. All SARS-CoV-2 pandemic strain sequences are available through GISAID (https://gisaid.org/). Additionally, a dataset of COVID antibodies is also available through CoV-AbDab (https://doi.org/10.1093/bioinformatics/btaa739). We could retrospectively evaluate the ‘Myopic’ case by using these antibodies. I.e. we can show that by setting up our simulator to a state equivalent to the beginning of the pandemic and simulating viral escape as a response to some of these real-world antibodies, the real viral strains observed later in the pandemic are ‘in distribution’ of our simulator’s outputs. That would show the simulator is ‘trustworthy’, and successfully running ADIOS in it would indicate real-world applicability.
>
> Going beyond simulation, we can validate ADIOS using bacteria phages, i.e. viruses that infect bacteria, and run shaping experiments in a wet lab with mutating bacteria and bacteria phages. Co-evolving bacteria phages (https://doi.org/10.1073/pnas.2104592118) would be a good real-world test ground for ADIOS, that is experimentally tractable and low-risk. ADIOS’s inner loop would correspond to bacteria phage evolving while the outer loop would be evolving the actual bacteria.
> ## 5. ADIOS for Cancer Therapy
> Monoclonal antibodies (mAb) are a common therapy used for cancer treatment. The way to adapt ADIOS to cancer therapy would be to still optimise antibodies in the outer loop, this time these would correspond to mAb antibodies. Instead of viral escape simulation in the inner loop, we would simulate the evolution of cancer cell’s growth factor receptors. The goal would be to design antibodies that shape cancer cells into cells that don’t proliferate well. We have added a short phrase to our manuscript suggesting this as a potential domain of application.
>
> Again, thank you for your detailed feedback! Hopefully, we have addressed all your concerns. If we did, could you consider updating your score for our paper?

---

> > ### Comment · Reviewer_kBGh · 2025-04-01
> >
> > ### On the dynamics
> > Although AF3 is a clear improvement over AF2 in terms of structure prediction, it still struggles with accurately modeling antibody-antigen complexes. More importantly, it doesn't capture true molecular dynamics—it simply samples a range of possible conformations, some of which may not reflect biologically relevant states. So while your proposed direction sounds promising, it may still be limited by the inherent shortcomings of AF3.

---

> > > ### Author Response · Authors · 2025-04-02
> > >
> > > ## On the Dynamics
> > > Thank you for your comment. We agree that AF3 doesn’t capture molecular dynamics, nor did we claim it does. Our previous response referred to “dynamics” resulting from antigen sequence mutation. This seems to be the result of a simple misunderstanding over what is meant by a “static antigen structure”. However, this is only relevant for future work.
> > > ## Rebuttal Acknowledgement
> > > In your original review, you list 4 weaknesses of the paper:
> > > 1. Testing on only one virus.
> > > 2. No experimental evidence for ADIOS for antimicrobial resistance and cancers.
> > > 3. No real-world validation.
> > > 4. Assumptions of static antigen structure.
> > >
> > > In our rebuttal, we’ve addressed all of them:
> > > 1. We run additional experiments on three other viruses. In all cases, ADIOS successfully achieves shaping and limits viral escape, see point 1 in our rebuttal.
> > > 2. We only briefly talk about antimicrobial resistance and cancer treatment in our paper, and we have now added further clarification regarding our claims, see point 5 in our rebuttal.
> > > 3. You already acknowledge the potential difficulty of real-world validation, and we have discussed at length how real-world validation of ADIOS is possible, see point 4 of our rebuttal. However, it’s beyond the scope of our current work, where the goal is to demonstrate that shaping with ADIOS is possible which we have successfully shown.
> > > 4. For this initial study assuming a static antigen structure was necessary, as modelling the changing antigen structure would hugely inflate the computational budget. Now that we have shown ADIOS works when Absolut! is the binding simulator we can build more accurate and expensive simulators to evaluate ADIOS as future work which we discuss in point 3 of our rebuttal.
> > >
> > > Given that you don’t provide any further concerns with our paper or rebuttal, we assume that our responses have satisfactorily addressed the weaknesses you initially highlighted. If so, we would greatly appreciate you reconsidering your score. Judging from your original review, it’s clear that you see a great deal of value in our work. However, we understand that a 'weak accept' was aligned with your initial critique. In light of our clarifications and extra experiments we conducted - including those with additional viruses - could you raise your overall score?

---

### Official Review · Reviewer_EjC1 · 2025-03-14

**Overall Recommendation:** 4

**Summary:**

The authors consider a very interesting and important problem of antibody development against viral strains which would control and defend against newer strains evolved from this one. So, the antibody development problem is viewed as a sequential decision making problem which is modeled as a two player zero-sum game between the antibody developer and the viral strain. The authors call their approach ADIOS (Antibody Development vIa Opponent Shaping) and the non-myopic antibodies they design as 'shapers'. They label their approach as a meta-learning problem where the outer loop is the antibody (shaper) design process and the inner one is the virus' adaptive (evolutionary) response. To demonstrate their approach, the authors build a simulator using the Absolut! framework.

**Update after rebuttal**
Based on the authors' responses to my review and the other reviews, I would like to retain my overall recommendation. I have highlighted my concerns and response to rebuttal in my comments.

**Claims And Evidence:**

The authors present a new approach for antibody design and demonstrate it using a simulated environment (based on work from literature). They do not make any theoretical claims.

**Essential References Not Discussed:**

I am not aware of any essential references that have been missed.

**Ethical Review Concerns:**

The authors already highlight potential concerns in their Impact Statement on Page 9.

**Experimental Designs Or Analyses:**

The authors perform several experiment using a simulator and also provide detailed analyses of their results.

**Methods And Evaluation Criteria:**

The authors do not use any benchmark datasets, but they use a simulator to show the efficacy of their approach. This simulator is based on other published work, but it would be good if the authors can add a note on the accuracy and real-word utility of this simulator. They do mention that their simulator is based on simplified models, but not its practical implication. Can they be used to generate a good proof of concept?

**Other Comments Or Suggestions:**

1. It would be good to describe the variables used in the algorithms where they are mentioned. For example $R_V$ in Algorithm 1, $F^H_{\hat v}$ in Algorithm 2.
2. It would be useful to the readers if the authors formally define the two player game including the various state, action and observation spaces, transition probabilities, observation functions, payoff functions, horizon and discounting used (if any).

**Other Strengths And Weaknesses:**

Strengths:
1. The paper is very well written and the authors' approach is clearly explained.
2. The approach presented by the author provides a way to try several other ideas from multi-objective RL and game theory in antibody design and thus is a useful research contribution.


Weaknesses:
1. Novelty: While the application is definitely novel to the best of my knowledge, the various components in the approach are taken from literature.

**Questions For Authors:**

1. How difficult is it to simulate virus evolution/mutation. Doesn't it depend on factors beyond antibodies, such as possibly another (non-human) host etc.?
2. Is it computationally involved to simulate viral evolution? Are the potential trajectories tractable?
3. What are the practical considerations involved in antibody design? Does the current approach account for effect of antibodies on human beings?
4. What are the time scales for evolution of viruses and development of new antibodies? How do these vary with respect to the disease progression timescale, severity/mortality rate of the disease, R0 etc.?
5. How accurate is the Absolut! framework based environment in estimating the binding strength of protein-protein interactions? Do these work for novel antibodies?
6.  When considering the potential harmful effects of future variants of viruses, are co-morbidities taken into account or just escape potential?
7. Is the action of the virus evolution and or antibody generation composes of generating a fixed-length amino-acid sequence at each time step or can this be a variable length sequence with some maximum length? Does protein folding play a role in designing feasible structures? The authors mention this as a future step in Section 7, but it would be interesting to note how will this affect the action space itself.
8. In equation (1), can there be multiple anti-targets for an antibody?
9. Have the authors explicitly defined a policy/strategy function, which is a mapping from the current state for the antibody design to a new antibody design?
10. Is some kind of an equilibrium reached by the authors' algorithm? Is an equilibrium expected? If so, what would be its characteristics?

**Relation To Broader Scientific Literature:**

This paper provides a novel way of considering antibody design (to the best of my knowledge), thereby providing interesting downstream research opportunities for the community.

**Theoretical Claims:**

There are no theoretical claims in this paper. The authors propose a new approach, which is actually a novel combination of existing approaches to a new problem.

---

> ### Author Rebuttal · Authors · 2025-03-31
>
> Thank you for your detailed review! We address everything possible within the character limit:
> ## Accuracy + Real-World Utility of Absolut!
> To address the real-world utility of our simulator we quote the original Absolut! paper: “Of note, Absolut! is neither suited nor designed to directly predict if and where an antibody sequence binds to an antigen in the real world. Rather, Absolut! has been developed with the premise that a successful ML antibody-binding strategy for experimental (real world) datasets should also perform well on synthetic datasets (and vice versa)”. Hence Absolut! is not designed for direct real-world predictions, but it has been shown that ML models which perform better according to Absolut! also perform well on real-world datasets. This makes Absolut! ideal as a testing ground for ADIOS. We reference this in Appendix Section C, where we explain Absolut! in more detail. To make this more clear, we can incorporate an additional explanation in the background section?
>
> ## Notation and MDP Definition
> We note that the functions $R_v$ and $F_v^H$ were not defined in the algorithms; we have therefore added their definitions. As for the formal definition of the MDP, we agree and have added a section detailing the MDP to the methods section.
> ## Answering Questions
> 1. Simulating viral evolution is generally challenging and computationally involved. However, recent works, such as EVEscape (https://doi.org/10.1038/s41586-023-06617-0) have had success forecasting viral mutations, so it is possible to have some predictive power, which is likely to increase as models and data improve. In our viral escape simulation, the virus responds to two main factors, the antibody and the viral target. In more accurate simulators you might want to account for other factors too, for example in real life the evolution will depend on: how chronic the virus is, what is the percentage of people that will be infected, and the mortality. To keep our approach computationally feasible we omit some of these factors.
> 2. See above.
> 3. The key consideration in antibody design is the specificity of the antibody, i.e. does it bind to the correct antigen and nothing else (https://doi.org/10.1186/s12929-019-0592-z). In ADIOS we just consider a region of the antibody sequence which is responsible for the specificity, the CDRH3 region. Designing the rest of the antibody sequence will affect many developability parameters in the human body such as PK, half-life, and effector functions, which are also important for antibody design but engineering these is independent of the specificity. Typically, the CDRH3s can be grafted to antibody scaffolds with good profiles of other properties, so our approach is suitable for the specificity question and selective pressure it creates.
> 4. The development of antibody therapeutics in the US on average takes about 8 years until the drug receives an FDA approval (https://doi.org/10.4161/mabs.2.6.13603). Viral evolution is very different between viruses, for example, RNA viruses (e.g. HIV, SARS-Cov2)  evolve faster than DNA viruses (e.g. Herpesvirus). In general, a higher mortality rate limits viral evolution and needs a faster immune response, while a higher R0 means more evolution opportunities for the virus given the higher number of infections.
> 5. See the section on Absolut! above.
> 6. In this work we mainly focus on viral escape potential and how easily targetable the virus is. However, in future work, with higher fidelity simulators we can model other factors such as mortality or co-morbidities.
> 7. In this work we assume a fixed sequence length for the antibody and the virus, we explain this in more detail in Section 4.1. An extension of this work where we allow insertions and deletions to the sequence is certainly possible, one way to adapt the action space to allow this would be to have a max sequence length for the insertions and an additional ‘None’ token for the deletions.
> Protein structure prediction models, such as AlphaFold3 (AF3), would be useful to validate the feasibility of structures resulting from the mutated sequences of antibodies and antigens. However, the action space would remain in the sequence space, and we could just include an extra ‘structure check’ of the sequence by passing it through AF3.
> 8. Yes, there can be multiple anti-targets in our model, but in our experiments, we stick to 1.
> 9. We don’t define the antibody’s policy function explicitly but we can easily add that if you think it’s helpful.
> 10. Figure 2b and Figure 2c show viral and antibody evolution plateauing. Eventually, an equilibrium must be reached, however, this may take an extremely long time. Do you think it would be interesting to run an ultra-long horizon experiment, to better understand the behaviour?
> ## Conclusion
> We appreciate the interest you have shown in our work! Is there anything we could change for you to raise your review to a “strong accept”? Also, any future work ideas would be great too!

---

> > ### Comment · Reviewer_EjC1 · 2025-04-09
> >
> > I thank the authors for the detailed response. They answer several of my questions. Regarding points about helpfulness of antibody policy definition and simulating equilibrium, I am not sure about this from a healthcare problem perspective; I just asked these from an RL perspective. Based on the authors' responses to my review and the other reviews, I would like to retain my overall recommendation.

---

> > > ### Author Response · Authors · 2025-04-09
> > >
> > > Thanks for your comment!
> > > Due to the character limit, we had to make our answers short and could not expand on your equilibrium question. We’ll take this opportunity to explain it a little further.
> > >
> > > To expand on the equilibrium question, in our current setting we model a one-shot deployment of an antibody therapy, reflecting the real-world limitations of releasing multiple rounds of therapies. Since we model the viral response to this single antibody, in the inner loop the virus will eventually converge to some local optima against the antibody. Technically, the virus will eventually converge to a distribution concentrated around the global optima (with the degree of concentration determined by the evolutionary parameters), but this will only happen in the limit. The antibody will converge to a different solution depending on the horizon it’s optimized for. In the extreme case as the horizon tends to infinity, and the virus is at its global optima, the optima choice of antibody coincides with the minimax regret choice.
> > >
> > > In the general setting, with new antibodies being deployed, the equilibrium will likely be a complicated dynamic equilibrium. Technically speaking, there is a weak pure Nash equilibrium where the virus exactly replicated the anti-target, $v = t_a^-$ and the antibody exactly replicated the target, $a = t_v^+$. To prove this, if $v = t_a^-$:
> > > $$
> > > R_a(v,a) = B(v,a) - B(t^-_a, a) - B(v, t^+_v) = B(t_a^-,a) - B(t^-_a, a) - B(t_a^-, t^+_v)  = - B(t_a^-, t^+_v)
> > > $$
> > > Which is a constant, satisfying the conditions for the antibody. For the virus, if $a = t_v^+$:
> > > $$
> > > R_v(v,a) = - R_a(v,a) = -B(v,a) + B(t^-_a, a) + B(v, t^+_v) = -B(v,t_v^+) + B(t^-_a, t_v^+) + B(v, t^+_v)  = B(t^-_a, t_v^+)
> > > $$
> > > Which is, again, a constant; completing the proof.
> > >
> > > This Nash equilibrium disappears once an additional anti-target is used. Since this is a weak Nash, the virus, being a naive evolutionary learner, has no capacity to remain at the equilibrium anyway. Hence, we hypothesise that the equilibrium will be dynamic.
> > >
> > > Part of the reason we have the extra target $t_v^+$ and anti-target $t_a^-$ is that without them a trivial pure-strategy Nash equilibrium exists.

---

### Official Review · Reviewer_NEd6 · 2025-03-14

**Overall Recommendation:** 3

**Summary:**

This paper introduces a long-term strategy using opponent shaping, a concept from game theory and reinforcement learning, to design antibodies that not only bind effectively to the virus but also influence the virus's evolutionary trajectory to make it less dangerous over time.

The algorithm involves three main components:
- Virus-Antibody Game: Models the interaction between the virus and the antibody as a two-player zero-sum game.
- Simulated Viral Escape: Simulates how the virus evolves to escape the antibody over time.
- Antibody Optimization: Optimizes the antibody to perform well against both the current virus and its future evolved variants.

**Claims And Evidence:**

Clams are clear.

**Essential References Not Discussed:**

The paper should also discuss more related work about antibody design such as Biological Sequence Design with GFlowNets, Reinforcement Learning for Sequence Design Leveraging Protein Language Models and more...

**Experimental Designs Or Analyses:**

I am not sure how Myopic Antibodies are generated. I believe we could have more baselines to be compared with. For example, PPO, rainbow DQN and even LOLA (https://arxiv.org/abs/1709.04326), AAA (https://arxiv.org/abs/2406.14662).
I also think since we already have some trajectories of the virus mutation. We can do back test on these trajectories and see if "simulates how the virus evolves to escape the antibody over time" really have a good prediction and the designed antibody would really perform good on the virus in the "future" in back test.

**Methods And Evaluation Criteria:**

Yes, they make sense.

**Other Comments Or Suggestions:**

I think more discussion about antibody design methods and more baseline would make the paper better. I am not sure how Myopic Antibodies are generated. I believe we could have more baselines to be compared with. For example, PPO, rainbow DQN and even LOLA (https://arxiv.org/abs/1709.04326), AAA (https://arxiv.org/abs/2406.14662).
I also think since we already have some trajectories of the virus mutation. We can do back test on these trajectories and see if "simulates how the virus evolves to escape the antibody over time" really have a good prediction and the designed antibody would really perform good on the virus in the "future" in back test.

**Other Strengths And Weaknesses:**

The paper is overall clear to me.

**Questions For Authors:**

See Other Comments Or Suggestions

**Relation To Broader Scientific Literature:**

Medical science, drug discovery.

**Theoretical Claims:**

No theoretical claims in this paper.

---

> ### Author Rebuttal · Authors · 2025-03-31
>
> Thank you for your review!
>
> As we understand, you have three primary concerns:
> 1. Lack of discussion of antibody design methods
> 2. Unclear how Myopic Antibodies are generated
> 3. No comparison to other reinforcement learning algorithms
> ## 1. Discussion of Antibody Design Methods
> You are completely correct, and we have updated the paper to include references to more antibody design methods. The reason we didn’t include such references originally is because ADIOS is somewhat agnostic to the choice of the antibody optimization algorithm used, the primary difference is that ADIOS emphasises the importance of accounting for the adaptation of the virus using simulated evolution. However, it is important to discuss the breadth of different antibody design methods in our related work section, which we have now rectified with additional references. In particular, we now reference energy-based antibody optimization methods (https://doi.org/10.1371/journal.pone.0105954, https://doi.org/10.1371/journal.pcbi.1006112), including optimization methods based on GFlowNets (https://doi.org/10.48550/arXiv.2411.13390), sequence-based language models (https://doi.org/10.1093/bioinformatics/btz895, https://doi.org/10.1038/s41598-021-85274-7) and structure-based approaches relying on GNNs (https://doi.org/10.48550/arXiv.2207.06616) and diffusion models (https://doi.org/10.48550/arXiv.2308.05027).
> ## 2. Generation of Myopic Antibodies
> The Myopic Antibodies are generated in the exact same way as the other shaper antibodies, but with a horizon of 0 in the objective. We now recognise that our explanation of this was unclear, as we didn’t explicitly state that we also optimize myopic antibodies in the same way we optimize shapers. We have now changed line 233 from “To optimise antibody shapers, we …” to “To optimise both shapers and myopic antibodies, we …” to rectify this. Thank you for bringing this to our attention!
> ## 3. Comparison to RL Baselines
> We don’t compare against several baselines because the core question of our current work is whether it is possible to apply opponent shaping to viruses and design effective antibody shapers, which we have now successfully demonstrated.
>
> However, your suggestion is a natural follow-up question - given that we can shape viral evolution, what algorithms (PPO, Rainbow DQN, etc.) allow us to create the optimal antibody shapers in the outer loop? This becomes significantly more viable to test due to our enormous speed up of Absolut! (x10,000), making fast, full training rollouts on the GPU possible. Still, there are some challenges in adapting a few of the works you mentioned to our problem setting. In particular, a lot of opponent shaping literature, such as LOLA and AAA, rely on assumptions suitable for RL-trained/humanlike agents, but less suitable to biological adaptive agents, such as viruses, which do not perform value iteration but evolve instead.
> ## Back-Testing on Historical Data
> Thank you for this suggestion. We fully agree with you! We are already planning on doing the back-testing, but it requires a much higher fidelity simulation of both binding and viral escape. As a result, we are working on extensions to this work, where instead of using Absolut! as our binding simulator we use more complex models. These include AlphaFold3 (and similar) to predict the changing structure of antibodies and viruses, as well as protein-protein docking models to estimate the binding strength based on the structure. Using other predictive models like EVEscape (https://doi.org/10.1038/s41586-023-06617-0), we also account for additional factors that influence viral escape. We can retrospectively evaluate the ‘Myopic’ case by using historical antibodies. I.e. we can show that by setting up our simulator to a state equivalent to the beginning of the pandemic and simulating viral escape as a response to some of these real-world antibodies, the real viral strains observed later in the pandemic are ‘in distribution’ of our simulator’s outputs. That would show the simulator is ‘trustworthy’, and successfully running ADIOS in it would indicate real-world applicability.
> ## Additional Experiments
> To better evaluate ADIOS we have now also conducted extra experiments with 3 additional viruses: flu, MERS and the West Nile virus. In all of these cases, described in the updated manuscript, our experiments show that the antibody shapers generated by ADIOS can successfully shape the viruses. The results of these experiments are now added to the paper as well.
> ## Conclusion
> Hopefully, we have addressed all your concerns! If we did, would you consider raising your score for our paper? If not, please let us know any further questions or suggestions. We are very focused on making this paper as high quality as possible, so we really appreciate your feedback.

---

> > ### Comment · Reviewer_NEd6 · 2025-04-04
> >
> > Hi,
> >
> > Thanks for the rebuttal. I still think reinforcement learning baselines are very important to this work, although you already show "it is possible to apply opponent shaping to viruses and design effective antibody shapers". I understand the time is limited to implement new experiments. I would maintain my score of the paper. Good luck!

---

> > > ### Author Response · Authors · 2025-04-08
> > >
> > > Thank you for your continued engagement with our work! We're glad to see that the RL baselines concern is your only remaining point, which suggests we've successfully addressed all your other feedback.
> > >
> > > If we understand correctly, you are suggesting using RL baselines in one of three possible places:
> > > 1. In the inner loop for the virus
> > > 2. In the outer loop, for shapers
> > > 3. For myopic antibodies
> > >
> > > Additionally, you also suggest testing ADIOS against other opponent shaping algorithms. Below we consider all of these points.
> > >
> > >
> > > ## 1. Inner loop RL baselines for the virus:
> > > We deliberately use an evolutionary algorithm to model viral evolution, reflecting biological processes of mutation and selection rather than an RL-based mechanism. Substituting an RL algorithm here would undermine this biological realism.
> > >
> > >
> > > ## 2. Outer loop RL baselines for shapers:
> > >  While we understand your suggestion to use algorithms like PPO or Rainbow DQN, implementing RL in the outer loop presents significant challenges. RL algorithms like PPO usually require 10^5+ rollouts to make policy improvements (https://doi.org/10.48550/arXiv.2005.12729). In our most expensive setting with horizon H=100, we use at most 12,000 rollouts. While applying RL to the outer loop is theoretically possible, it would require significantly redefining the MDP and making substantial adaptations to work effectively within these constraints. These modifications would be extensive enough that we believe they merit exploration in a separate paper rather than within this initial work on ADIOS.
> > >
> > > We also appeal to the opponent shaping literature. Model-Free Opponent Shaping (M-FOS,
> > > https://doi.org/10.48550/arXiv.2205.01447) simply chooses one effective outer-loop optimizer (either Genetic Algorithms or PPO) to demonstrate the viability of shaping. Similarly, we believe additional RL algorithms are not essential to validate ADIOS, as a genetic algorithm is sufficient.
> > >
> > > ## 3. RL baselines for myopic antibodies:
> > > In our paper, the myopic antibody represents the baseline, equivalent to the ‘naive learner’ in other opponent shaping literature such as LOLA or M-FOS. Although a more sophisticated myopic optimizer might exist, Figure 2a demonstrates that optimizing for the myopic objective doesn’t improve performance on the true objective. Figure 2d similarly shows the myopic strategy plateauing. We therefore expect that even an RL-based approach to the myopic objective would not notably improve its true-objective performance.
> > >
> > > ## 4. Opponent shaping baselines for ADIOS:
> > > Standard opponent shaping evaluation includes comparing against both other shaping algorithms and naive learners. We include the latter through our myopic antibody baseline. Most shaping algorithms are unsuitable comparisons as they assume reinforcement learning-based opponents, not evolutionary dynamics. M-FOS could potentially work, but ADIOS is already an adaptation of M-FOS to this biological context, making such comparison redundant.
> > >
> > > ## Conclusion
> > > Overall, we believe our comparison of shaper vs. myopic antibodies adequately demonstrates ADIOS’s key contribution: how shaping can influence viral evolution. Our additional tests with multiple viruses further reinforce this finding.
> > > We trust this clarifies why we have not included RL-based or other opponent shaping baselines and why we see these as future research directions. Thank you for your thoughtful feedback and for considering this explanation in your final assessment.

---

### Official Review · Reviewer_Y7Ln · 2025-03-14

**Overall Recommendation:** 2

**Summary:**

The manuscript presents ADIOS as a method for optimizing antibodies while considering viral escape. In essence, the proposed approach is a simplified version of adversarial training. Experiments were conducted using a binding prediction method (Absolut!), and the results indicate that the proposed method outperforms the one that does not account for viral escape.

**Claims And Evidence:**

One major concern with the method is that the optimization results are evaluated using Absolut!, while the proposed method may generate data that lies outside of Absolut!’s distribution. Therefore, an improvement in the results measured by Absolut! does not necessarily reflect an actual enhancement in the final antibody. It is highly likely that the resulting sequences fall outside the distribution of Absolut!, which is, in fact, a key challenge in this field. As an application-oriented paper, such considerations should be addressed. Thus, the results presented in the manuscript cannot conclusively demonstrate that the proposed method leads to antibodies with improved properties in real-world scenarios.

**Essential References Not Discussed:**

No

**Experimental Designs Or Analyses:**

I have reviewed all the experimental designs, and I found issues with both the evaluation metrics (the computed binding scores) and the data (only one type of antigen was used). These limitations prevent the proposed method from demonstrating its performance and generalizability in real-world scenarios.

**Methods And Evaluation Criteria:**

The performance of the proposed method is evaluated using Absolut!, which does not demonstrate the method's applicability in real-world scenarios. Additionally, the proposed method is tested with only one type of antigen, making it impossible to conclude that the method would be effective for other antigens.

**Other Comments Or Suggestions:**

No other comments

**Other Strengths And Weaknesses:**

Strengths: The acceleration of Absolut! contributes to the broader user community of Absolut!.

Weaknesses:

1. The novelty of the algorithm is limited.

2. In terms of paper presentation, the figures in the supplementary material should be renumbered rather than continuing the numbering from the main text.

**Questions For Authors:**

1. What is the computational speed of the proposed method?
2. How is the correctness of the "binding" prediction determined when binding poses change?

**Relation To Broader Scientific Literature:**

Antibody optimization is a critical area of research, with significant implications for drug and vaccine design, while viral escape remains a major challenge in vaccine development. Numerous efforts have been made to build antibody binding prediction models, but none have achieved perfect accuracy. Although the motivation behind this paper is sound, the method and experimental design fail to address the key challenges encountered in real-world applications.

**Theoretical Claims:**

There's no theoretical claims in this manuscript.

---

> ### Author Rebuttal · Authors · 2025-03-31
>
> Thank you for your review. Below we address your concerns and questions:
> # Extra Experiments
> Following your feedback, we have now conducted extra experiments with 3 additional viruses: flu, MERS and the West Nile virus. In all of these cases, our experiments show that the antibody shapers generated by ADIOS successfully shape the viruses, limiting viral escape. We have now added these results to the paper, alongside correcting the enumeration of appendix figures.
> # Sampling Outside of Absolut!’s Distribution
> Thank you for bringing up the important problem of a binding model's distribution, or in other words its domain of applicability. We have carefully considered this problem and, in fact, one of the main reasons we decided to use Absolut! as our binding simulator is to avoid sampling outside of the distribution.
>
> As we explain in Section 3 of our paper, sequence-based ML models like Mason et al., 2021 (https://doi.org/10.1038/s41551-021-00699-9); Lim et al., 2022 (https://doi.org/10.1080/19420862.2022.2069075); Yan Huang et al., 2022 (https://doi.org/10.3389/fimmu.2022.1053617), as data-driven predictive models struggle to generalise beyond their training distribution and are therefore not suitable for our application which explores novel viral mutations.
>
> However, Absolut! is a mechanistic model (not data-driven) that we use as an oracle for querying the binding strength of new sequences. As we write in Section 3, “For any antibody-antigen pair, Absolut! enumerates possible binding poses and computes their energy using the Miyazawa-Jernigan potential” This process doesn’t have a training distribution or a domain of applicability and is well suited to our application which requires mutating both the virus (antigen) and the antibody.
> # Real-World Applicability
> We appreciate the challenges of binding prediction, and acknowledge that using Absolut! (or any other simulator) comes at a cost of real-world applicability. However, it has been shown that ML models that perform better on Absolut! perform well on real-world datasets too (https://doi.org/10.1038/s43588-022-00372-4). Besides, using more realistic models is extremely expensive and cheaper data-driven models are not suitable due to their limited domain, making Absolut! an ideal choice for the current work.
>
> More broadly, breakthroughs in biology have come from building on previous computational works with initially limited real-world applicability. For example, the development of effective mRNA vaccines has been credited at least in part to RNA Folding algorithm. Original papers such as Nussinov and Jacobson, 1980 (https://doi.org/10.1073/pnas.77.11.6309) and Zucker and Stiegler, 1981 (https://doi.org/10.1093/nar/9.1.133) laid the foundations to more modern heuristics for RNA structure prediction and therefore therapeutics applications.
>
> We are currently working on extensions to this work,  where instead of using Absolut! as our binding simulator we use more complex models. These include AlphaFold3 (and similar), to predict the changing structure of antibodies and viruses, as well as protein-protein docking models to estimate the binding strength based on the structure. We also account for additional factors that influence viral escape. However, Absolut! is a necessary first step to prove that ADIOS is feasible and effective. Due to its speed, it is also a great testbed for developing methods which can then be scaled to more computationally expensive models.
> # Answers to Questions
> 1. In Figure 2d, we show the number of binding samples required to achieve a given antibody fitness for different shaping horizons. In Table 1, we report how long a single binding sample takes in our GPU-accelerated implementation: $2.1 ×10^{−4}s$. As the binding calculation is the main computational cost, a single run takes up to 4000s ~ 1h. For Figure 2d we chose to show binding samples on the x-axis (instead of runtime) because it provides a reference point for future work where one will want to use a more expensive (i.e. slower) binding simulator. Would it be helpful to discuss in the paper how binding calculation is the main computational cost? Or show a plot with the computational speed of our method (on our hardware) on the x-axis?
> 2. Our binding simulator, Absolut!, is not a pre-trained predictive model but a mechanistic one instead. Absolut! applies the same scoring computation to any possible binding pose to calculate the binding energy, a priori, there is no one pose where the binding calculation is systematically more “correct”. Thus, we hypothesise that mean correctness of the binding score should not systematically change as we mutate the sequences and/or change binding poses.
> # Conclusion
> We value and respect your critique, but we believe it is in large part due to a misunderstanding of Absolut!. If you disagree, please let us know why! We are always looking to improve the quality of our work. Otherwise, we would appreciate a re-evaluation of your review score.

---

### Decision · Program_Chairs · 2025-05-01

**Decision:**

Accept (poster)

**Comment:**

The paper "ADIOS: Antibody Development via Opponent Shaping" presents a novel approach to designing antibody therapies that not only defend against current viral strains but also shape viral evolution toward less harmful variants. This is achieved through a meta-learning framework involving viral evolution simulation and antibody optimisation. The method's novelty lies in its game-theoretic approach combined with mechanistic modelling using Absolut!. The Absolut! software from Victor Greiff's lab takes PDB antigens, converts them into lattice representation with integer positions together with CDR3 Amino Acid sequences, and computes their best binding around a lattice antigen, and then generates features of antibody-antigen bindings directly usable for downstream ML such as in this paper. Notably Absolut! is a mechanistic model and not an ML model, which seemed to cause some confusion with one reviewer in particular. The work is significant in enhancing the understanding of long-term immune strategies, with potential implications for antimicrobial resistance and cancer treatment.

The paper received mixed reviews initially, with scores of 2, 3, and 4. Reviewers highlighted the rigorous methodology and potential significance of the work, though they expressed concerns regarding the real-world applicability and comparisons with existing methods. In the discussion, the authors provided additional experimental data, addressed misunderstandings related to the Absolut! framework, and conducted extra experiments on three more viruses. This effort improved the perception of the paper, with reviewers appreciating the clarifications and additional data, though they maintained their initial scores (the reviewer who gave the lowest score didn't respond to the rebuttal).

The primary strengths of this submission are its novel integration of computational techniques to address viral evolution and its empirical validation through simulations. However, the limitations include reliance on simulation models for validation and the lack of real-world experimental data. The paper's clarity and logical structure make it accessible to both computational and biological audiences, although some points, like the role of Absolut!, initially required deeper explanation to avoid reviewer misconceptions.

Although the work's real-world impact is yet to be fully determined, the novel computational approach and comprehensive feedback integration indicate significant academic merit. Therefore, I recommend accepting this paper as a valuable contribution to ICML.